# Human neurocomputational mechanisms of guilt-driven and shame-driven altruistic behavior

Ruida Zhu[1], Huanqing Wang[2], Chunliang Feng[3], Linyuan Yin[1], Ran Zhang[4,5], Yi Zeng[5,6,7,8,9], Chao Liu[4,5]*

[1]Department of Psychology, Sun Yat-sen University, Guangzhou, China; [2]Department of Psychology, The Ohio State University, Columbus, United States; [3]Key Laboratory of Brain, Cognition and Education Sciences, Ministry of Education, School of Psychology, Center for Studies of Psychological Application, Guangdong Key Laboratory of Mental Health and Cognitive Science, South China Normal University, Guangzhou, China; [4]State Key Laboratory of Cognitive Neuroscience and Learning, and IDG/McGovern Institute for Brain Research, Beijing Normal University, Beijing, China; [5]Beijing Key Laboratory of Safe AI and Superalignment, Beijing, China; [6]Beijing Institute of AI Safety and Governance, Beijing, China; [7]Brain-inspired Cognitive AI Lab, Institute of Automation, Chinese Academy of Sciences, Beijing, China; [8]University of Chinese Academy of Sciences, Beijing, China; [9]Long-term AI, Beijing, China

*For correspondence:
liuchao@bnu.edu.cn

## eLife Assessment

This is an **important** study on how dissociable emotions of shame and guilt emerge from cognitive processes and guide behavioral responses. The task is well designed and yields **compelling** behavioral, computational, and neural evidence elucidating the cognitive link between emotions and compensatory decisions. The work has broad theoretical and practical implications across a range of disciplines concerned with human behavior, including psychology, neuroscience, economics, public policy, and psychiatry.

**Abstract** Although prior research has examined the psychological and neural correlates of guilt and shame, the cognitive antecedents that trigger them, as well as their transformation into social behavior, remain insufficiently understood. We developed a novel task to investigate how two cognitive antecedents, harm and responsibility, elicit guilt and shame, and how these emotions subsequently drive compensatory behavior, by combining functional magnetic resonance imaging (fMRI) with computational modeling in human participants. Behaviorally, we found that harm had a stronger impact on guilt, whereas responsibility had a stronger impact on shame. Moreover, compared to shame, guilt exerted a greater effect on compensation. Computational modeling results indicated that the integration of harm and responsibility by individuals is consistent with the phenomenon of responsibility diffusion. The fMRI results revealed that brain regions associated with inequity representation (posterior insula) and value computation (striatum) encode this integrated measure. Individual differences in responsibility-driven shame sensitivity were associated with activity in theory-of-mind regions (e.g. temporoparietal junction). Guilt-driven and shame-driven compensatory behavior recruited distinct neural substrates, with shame-driven compensatory sensitivity being more strongly linked to activity in the lateral prefrontal cortex, a region implicated in cognitive control. Our findings provide computational, algorithmic, and neural accounts of guilt and shame.

## Introduction

Guilt and shame, two prominent moral emotions, underpin a wide array of social behaviors and phenomena, including norm compliance, cooperation, face-saving strategies, suicides, and even large-scale conflicts like wars (*Haidt, 2003*; *Sznycer et al., 2021*). Although guilt and shame often coexist after moral violation and serve to inhibit further transgressions (*Eisenberg, 2000*; *Lewis, 1971*), they differ in their associations with psychological and behavioral issues (*Tangney and Dearing, 2003*). For instance, shame is commonly associated with higher levels of anxiety, depression, stress, eating disorders, and aggression, whereas guilt is typically unrelated or negatively related to these issues (*Blythin et al., 2020*; *Caldwell et al., 2021*; *Cândea and Szentagotai-Tătar, 2018*; *Elison et al., 2014*; *Kim et al., 2011*; *Schuster et al., 2021*).

Recognizing their distinctions, a growing body of research in psychology and neuroscience has sought to elucidate the psychological and neural mechanisms differentiating guilt and shame, providing insights into how these emotions are processed and regulated. Psychological studies have revealed that guilt involves more concerns on one's actions on others (i.e. negative behavioral impacts), whereas shame involves more concerns on one's self-image (i.e. negative self-evaluations; *Lewis, 1971*; *Tangney et al., 2007*; *Tangney and Dearing, 2003*). Echoing and complementing these findings, neuroscience research showed that neural activity in brain regions related to other-oriented theory-of-mind processing (temporoparietal junction [TPJ] and dorsomedial prefrontal cortex [DMPFC]), self-oriented self-referential processing (anterior cingulate cortex [ACC] and DMPFC), and cognitive control (lateral prefrontal cortex [LPFC]) can distinguish guilt and shame (*Piretti et al., 2023*; *Zhu et al., 2019a*).

Although the psychological processes and neural activities related to guilt and shame experience are relatively well-documented, the cognitive antecedents of these emotions and their neural representation remain insufficiently understood. Existing research has identified harm (i.e. severity of harm) and responsibility (i.e. responsibility for harm) as cognitive antecedents influencing guilt (*Abrams and Doosje, 2011*; *Berndsen et al., 2004*; *Cehajić-Clancy et al., 2011*; *Gao et al., 2021*; *Iyer et al., 2007*; *Koban et al., 2013*; *Li et al., 2020*; *Tangney, 1992*; *Yu et al., 2014*) and shame (*Iyer et al., 2007*; *Koban et al., 2013*; *Tangney, 1992*). However, it remains unclear whether these factors differ in the strength of their influence on guilt and shame. Functionalist theories propose that guilt functions to minimize and repair undue harm on valued others, thereby addressing the adaptive problem of insufficiently valuing others—a behavior that can indirectly harm oneself (e.g. harming one's partner can disrupt cooperation, ultimately damaging one's own interests; *Sznycer, 2019*). In contrast, shame functions to prevent and mitigate the threat of being devalued by others, thus addressing the adaptive problem of reputational damage to the self (e.g. being perceived as deficient in competence or morality by others; *Landers et al., 2024a*; *Sznycer, 2019*; *Sznycer et al., 2021*). If the functionalist theories are correct, the intensity of guilt and shame should more closely track the information that reflects the adaptive problems they are meant to address (e.g. *Sznycer et al., 2016*). Accordingly, severity of harm—a factor reflecting inadequate valuation of others—is expected to have a stronger impact on guilt than on shame, whereas responsibility for harm—a factor tied to devaluation by others, particularly in open and transparent social contexts—is anticipated to exert a greater influence on shame than guilt (see Appendix 1 for illustrative examples).

Previous neuroscience studies have examined the neural response to harm or responsibility. Greater harm inflicted on others is associated with stronger activation in the DMPFC, TPJ, and insula (*Crockett et al., 2017*; *Koban et al., 2013*). Greater responsibility for harm is linked to stronger activation in the anterior middle cingulate cortex (aMCC) (*Li et al., 2020*; *Yu et al., 2014*). Despite these findings, it remains elusive how the human brain integrates harm and responsibility when both factors are simultaneously present and dynamically vary. One possibility is that distinct brain regions separately encode harm and responsibility, with their signals converging in other regions responsible for integrating these two factors (e.g. *Hu et al., 2017*; *Yu et al., 2018*). Alternatively, some brain regions may directly encode an integrated representation of harm and responsibility, potentially through computational processes such as their product or quotient (e.g. *Gray et al., 2002*).

According to appraisal theory, emotions are not directly elicited by cognitive antecedents (e.g. harm and responsibility) but arise from the appraisal processes applied to these antecedents (*Lazarus and Smith, 1988*; *Moors et al., 2013*). Since appraisal processes vary across individuals, the same stimuli can evoke emotions of distinct intensities and even distinct types across different individuals

(*Ellsworth, 2013*; *Moors et al., 2013*). Individual differences in emotional experiences, such as guilt and shame, may originate from the variations in neural responses to their cognitive antecedents (*Hamann and Canli, 2004*; *Morawetz and Basten, 2024*). However, the neural substrates that determine the extent to which harm and responsibility are transformed into guilt and shame remain largely unknown.

Shifting the focus from the origins of guilt and shame to their consequences, existing evidence on the distinct associations between these emotions and behavior remains incomplete. Guilt is widely recognized as a powerful motivator of altruistic behaviors, such as offering apologies and making amends (*Graton and Ric, 2017*; *Howell et al., 2012*; *Ketelaar and Tung Au, 2003*; *Zhu et al., 2017*). In comparison, shame appears to have a weaker promotive effect on altruistic behaviors compared to guilt (*de Hooge et al., 2008*; *de Hooge et al., 2007*; *Declerck et al., 2014*) and is commonly associated with non-cooperative and antisocial behaviors, including hiding, evasion, self-improvement, externalizing blame, and aggression (*de Hooge et al., 2010*; *Gausel and Leach, 2011*; *Landers et al., 2024a*; *Tangney et al., 1996*; *Zhu et al., 2019c*). Most studies examined only guilt or only shame, restricting direct comparisons of their behavioral effects. The few studies that have directly compared these emotions relied on participants' recollections of past personal events (*de Hooge et al., 2007*) or imagined scenarios (*Ghorbani et al., 2013*) to elicit guilt and shame. Variations in recollections and imagined contexts may introduce potential confounds, making it difficult to discern whether the observed behavioral differences are caused by guilt and shame or by irrelevant contextual factors.

Neurally, whether the transformation of guilt and shame into behavior depends on distinct neural bases remains an open question. *Yu et al., 2014* found that the aMCC and midbrain nuclei are involved in linking guilt to compensatory behavior. To the best of our knowledge, the neural bases underlying the conversion of shame into behavior have yet to be explored. Naturally, this also means that no studies have directly compared the neural correlates of guilt-driven and shame-driven behavior.

To address the research gaps outlined above and provide computational, algorithmic, and neural accounts of guilt and shame (see *Yu et al., 2024*), we combined computational modeling and functional magnetic resonance imaging (fMRI) with a novel interpersonal game. In this game, we independently manipulated the harm inflicted on a victim and the responsibility of participants. Throughout the experiment, participants engaged in compensatory decision-making while undergoing fMRI scanning, followed by post-experimental self-reports of their guilt and shame feelings. This paradigm allowed us to explore the associations among harm, responsibility, guilt, shame, and compensation, as well as to uncover relevant neural substrates. Based on the functionalist theories (*Baumeister et al., 1994*; *Gilbert, 1997*; *Landers et al., 2024a*; *Sznycer, 2019*; *Sznycer et al., 2016*), we predicted that harm

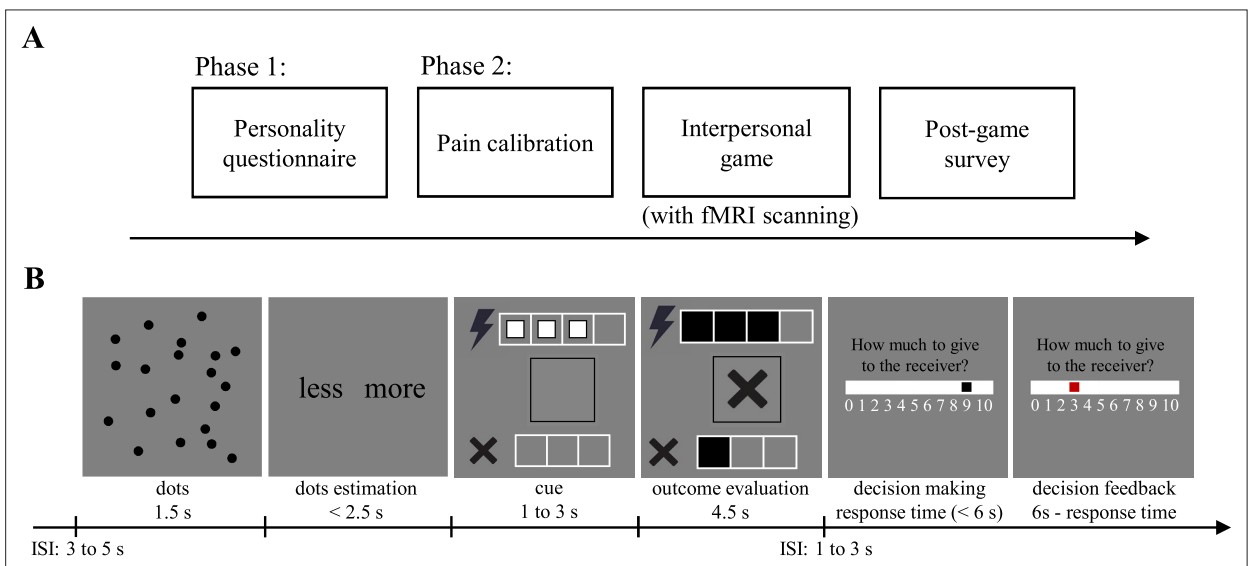

**Figure 1.** Experimental procedure. (**A**) Timeline for the whole experiment. (**B**) Timeline for the interpersonal game. In this example, the participant, who was one of the four deciders, made an incorrect estimation, as did one of the other deciders. The participants subsequently decided to allocate three monetary tokens to the receiver. ISI, inter-stimulus interval.

would have a greater impact on guilt than shame, whereas responsibility has a greater impact on shame than guilt. Based on previous research on the relationships between guilt, shame, and altruistic behavior (*de Hooge et al., 2007*), we predicted that guilt would exert stronger influence on compensation than shame. Drawing from existing knowledge of the neural correlates of guilt and shame (*Bastin et al., 2016*; *Li et al., 2020*; *Yu et al., 2020*; *Zhu et al., 2019a*), we predicted that brain regions involved in emotional processing (e.g. insula), self-referential processing (e.g. DMPFC), theory-of-mind processing (e.g. TPJ), and cognitive control (e.g. LPFC) would play important roles in the neural representation of harm and responsibility, and the emergence of guilt-driven and shame-driven compensation.

## Results

The experiment consisted of two phases (*Figure 1A*). In the first phase, participants completed online personality questionnaires assessing trait guilt, trait shame, trait gratitude, and social value orientation (SVO) at least one day before the laboratory session. In the second phase, each participant came to the laboratory individually. They first underwent an individualized pain calibration procedure designed to familiarize them with the different levels of electric shock used later in the interpersonal game and then completed the interpersonal game while undergoing fMRI scanning (*Figure 1B*). In the interpersonal game, the participant acted as one of four deciders performing a dots estimation task. If any decider made an incorrect estimate, a receiver received a painful electric shock whose intensity was randomly determined. The task was repeated across multiple trials. Unbeknownst to the participant, the other deciders were confederates and the receiver was fictitious; all outcomes were predetermined. After each outcome, the participant decided how much financial compensation to provide to the receiver. We focused on trials in which the participant, acting as a decider, was informed that they had made an incorrect estimate. Harm (levels 1–4) was manipulated through the intensity of the electric shock, whereas responsibility (levels 1–4) was manipulated by varying how many of the other deciders also made incorrect estimates. Following the game, participants completed a post-game survey in which they reported their feelings of guilt and shame for different outcomes, rated their perceived responsibility, and answered additional questions. Further procedural details are provided in the Materials and methods section.

### Manipulation checks

Participants' pain ratings differed significantly across the four harm levels (harm level 1 vs. harm level 2: $F(1,41) = 324.63$, p<0.001, partial $\eta^2$=0.888; harm level 2 vs. harm level 3: $F(1,41) = 181.26$, p<0.001, partial $\eta^2$=0.816; harm level 3 vs. harm level 4 $F(1,41) = 247.41$, p<0.001, partial $\eta^2$=0.858). Their responsibility ratings differed significantly across the four responsibility levels (responsibility level 1 vs. responsibility level 2: $F(1,41) = 36.41$, p<0.001, partial $\eta^2$=0.470; responsibility level 2 vs. responsibility level 3: $F(1,41) = 46.47$, p<0.001, partial $\eta^2$=0.531; responsibility level 3 vs. responsibility level 4: $F(1,41) = 139.24$, p<0.001, partial $\eta^2$=0.773) (*Appendix 2—figure 1*, *Appendix 3—table 1* and *Appendix 3—table 2*). In addition, participants reported higher responsibility ratings when they made incorrect estimates compared to correct estimates (making right estimates vs. responsibility level 1: $F(1,41) = 56.63$, p<0.001, partial $\eta^2$=0.580; making right estimates vs. responsibility level 2: $F(1,41) = 148.25$, p<0.001, partial $\eta^2$=0.783; making right estimates vs. responsibility level 3: $F(1,41) = 237.52$, p<0.001, partial $\eta^2$=0.853; making right estimates vs. responsibility level 4: $F(1,41) = 495.73$, p<0.001, partial $\eta^2$=0.924). These results suggest our manipulations of harm and responsibility were successful.

### Distinct impacts of harm and responsibility on guilt and shame

To confirm and compare the effects of harm and responsibility on guilt and shame, we ran two linear mixed-effect regression analyses. In the first analysis, participants' emotion ratings were regressed onto harm levels, emotion types (guilt vs. shame), and their interaction. In the second analysis, participants' emotion ratings were regressed onto responsibility levels, emotion types (guilt vs. shame), and their interaction. To determine whether harm and responsibility had significant effects on guilt and shame, respectively, we performed simple slope analyses for each emotion type. We found that harm significantly increased participants' guilt ratings ($\beta_{guilt} = 0.74$, $T(67) = 8.92$, p<0.001) and shame ratings

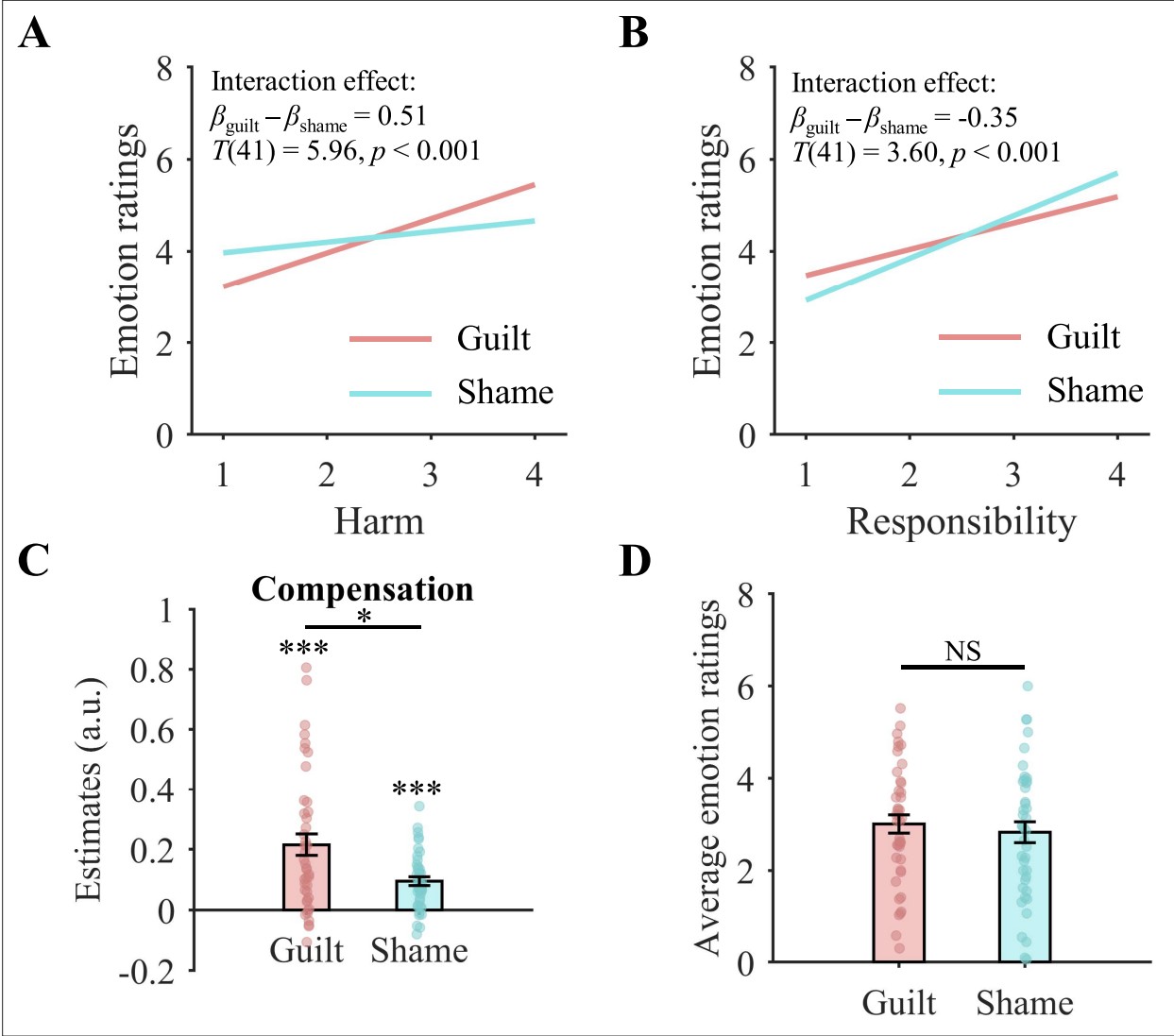

**Figure 2.** Behavioral results. (**A**) Harm had a stronger effect on guilt than on shame. (**B**) Responsibility revealed a stronger effect on shame than on guilt. (**A, B**) We created regression equations for guilt and shame ratings using the coefficient estimates from the linear mixed-effect regression analyses. To illustrate the impacts of harm and responsibility, we visualized the regression lines based on these equations. (**C**) The coefficient estimates from the linear mixed-effect regression model showed that, compared with shame, guilt exerted a larger effect on compensation. (**D**) Participants' average guilt and shame feelings showed no significant difference. (**C, D**) Data are shown as the mean ± standard error with overlaid dot plots. $^*p < 0.05$, $^{***}p < 0.001$; NS, not significant.

($\beta_{shame} = 0.23$, $T(67) = 2.77$, p=0.007). Similarly, responsibility significantly increased participants' guilt ratings ($\beta_{guilt} = 0.57$, $T(61) = 5.27$, p<0.001) and shame ratings ($\beta_{shame} = 0.93$, $T(61) = 8.54$, p<0.001). Of importance, harm had a stronger effect on guilt than on shame, as indicated by the interaction effect between harm and emotion type ($\beta_{guilt} - \beta_{shame} = 0.51$, $T(41) = 5.96$, p<0.001; *Figure 2A*; *Appendix 3—table 3*). Conversely, responsibility exerted a greater influence on shame than on guilt, as shown by the interaction effect between responsibility and emotion type ($\beta_{guilt} - \beta_{shame} = -0.35$, $T(41) = 3.60$, p<0.001; *Figure 2B*; *Appendix 3—table 4*). To control for the potential influence of other interaction effects, we conducted an additional linear mixed-effect regression. We regressed participants' emotion ratings onto harm levels, responsibility levels, emotion types (guilt vs. shame), and all their interactions. The results showed that the interaction between harm and emotion type, as well as the interaction between responsibility and emotion type, remained significant (*Appendix 3—table 5*). Together, these findings demonstrate the differential impacts of harm and responsibility on guilt and shame.

## Guilt and shame differed in their effects on compensatory behavior

To investigate whether and how guilt and shame contributed to compensation, we tested four linear mixed-effect regression models. The models regressed the number of tokens that participants distributed to the receiver on different fixed-effect regressors (see Materials and methods). The results of the model comparison showed that the best model (Model III, *Appendix 3—table 6*) included guilt and shame ratings (but not their interaction) as the regressors. The statistical results of the best model revealed that both guilt ($\beta$=0.22, $T$(41) = 5.16, p<0.001) and shame ($\beta$=0.10, $T$(36) = 3.69, p<0.001) were significantly associated with compensation (*Appendix 3—table 7*). Furthermore, a direct comparison between the coefficient estimates indicated that guilt had a stronger effect on compensation than shame ($c^2$(1)=6.48, p=0.011; *Figure 2C*). This difference in the effects of guilt and shame on compensation was not attributable to differences in the intensity of guilt and shame, as participants' average guilt and shame ratings across trials showed no significant difference ($F$(1,41) = 1.11, p=0.297, partial $\eta^2$=0.026; *Figure 2D*). Our results provide solid evidence supporting that guilt, compared to shame, is more effectively converted into compensation.

## Personality traits and compensatory behavior

Consistent with previous findings (e.g. *Cohen et al., 2011*), we found that two dimensions of trait guilt scores were significantly correlated to participants' compensatory behavior (negative behavior-evaluations: $R$=0.39, p=0.010; repair action tendencies: $R$=0.33, p=0.030), whereas two dimensions of trait shame scores were not (negative self-evaluations: $R$=0.20, p=0.213; withdrawal action tendencies: $R$=0.16, p=0.315). These findings again demonstrate the distinct behavioral impacts of guilt and shame. The trait gratitude scores and SVO scores were not significantly correlated with compensatory behavior (trait gratitude: $R$=0.18, p=0.255; SVO: $R$=0.24, p=0.121).

## Computational models of compensatory behavior

After confirming their associations through separate linear mixed-effect regression analyses, we used utility models to characterize how individuals integrate harm with responsibility during guilt-driven and shame-driven compensatory decision-making within a unified computational framework. We constructed eight models across two model families, which aimed at capturing individuals' distinct latent psychological processes. They varied in whether self-interest was considered, whether compensatory baseline existed, and how harm and responsibility were integrated (see Materials and methods). Based on the Bayesian information criteria, Model 1.3 outperformed other alternative models (*Appendix 3—table 8*). This model assumes that, during compensatory decision-making, individuals disregard their self-interest, adopt a compensatory baseline, and integrate harm and the number of wrongdoers in the form of a quotient (rather than integrating harm and responsibility in the form of a product). The utility ($U$) of a compensatory decision is described as follows (Model 1.3):

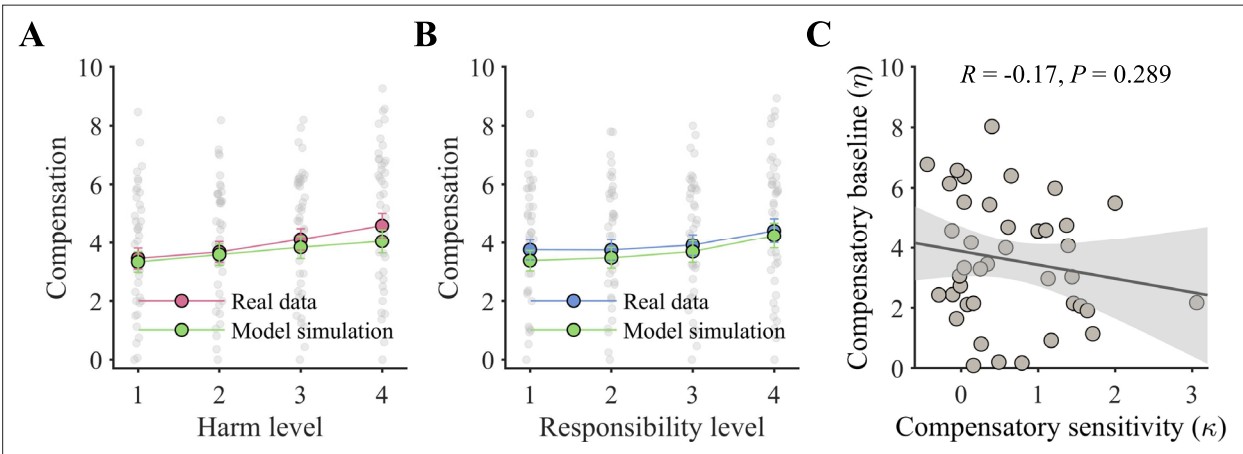

**Figure 3.** Computational modeling results. (**A, B**) Model simulations reproduced the behavioral patterns of compensatory decisions as influenced by harm (**A**) and responsibility (**B**). Data are shown as the mean ± standard error with overlaid dot plots. (**C**) The compensatory sensitivity ($\kappa$) and compensatory baseline ($\eta$) had no significant correlation. The line represents the least squares fit with shading showing the 95% confidence interval.

$$U\left(D\right) = -|\kappa \times \frac{H}{W} + \eta - D|$$

where $H$ and $W$ are the level of harm and the number of wrongdoers, respectively. $\frac{H}{W}$ represents the average harm for which participants are responsible. The parameter $\kappa$ reflects compensatory sensitivity. Higher $\kappa$ values indicate that participants are inclined to spend more tokens to compensate for the harm they are responsible for. The parameter $\eta$ denotes compensatory baseline. Higher $\eta$ values indicate that participants believe they should distribute more tokens to the receiver for compensation, regardless of the level of harm and the number of wrongdoers. $D$ is the number of tokens distributed to the receiver by participants. It is assumed that participants are averse to giving the receiver less compensation (undercompensation) and more compensation (overcompensation) they believe the receiver deserves.

We validated the winning model (Model 1.3) through a variety of analyses. Simulation tests indicated that the model could replicate the influence of harm level and responsibility level on compensation through simulation (*Figure 3A and B*) and predict participants' compensatory decisions with significantly higher accuracy than the chance level (mean accuracy: 31%, 95% confidence interval (CI): [27%, 36%], chance level: 9%). Parameter recovery tests revealed that the parameters were highly identifiable (Pearson correlation between real and recovered parameters: $r_\kappa$=0.95, CI = [0.95, 0.96]; $r_\eta$=0.99, CI = [0.99, 0.99]). No evidence supports that the psychological processes represented by the parameters $\kappa$ and $\eta$ overlap, as the Pearson correlations between the two parameters were not significant ($R$=–0.17, p=0.289; *Figure 3C*).

## Neural representation of cognitive antecedents

To investigate the neural representation of the cognitive antecedents of guilt and shame during outcome evaluation, we constructed general linear models (GLMs) to identify brain regions that respond parametrically to harm, the number of wrongdoers, the quotient of harm divided by the number of wrongdoers, and the product of harm and responsibility (responsibility = 5 – the number of wrongdoers; see Materials and methods). No brain region showed significant responses to harm, which implies the brain doesn't represent harm in isolation. The precentral and postcentral cortices, known to be involved in movement preparation (*Porro et al., 1996*; *Thoenissen et al., 2002*), exhibited positive parametric responses to the number of wrongdoers, potentially reflecting motor preparation for subsequent button-press action related to compensation (*Appendix 3—table 9*). The results of the neural representation of the level of harm and the number of wrongdoers remained consistent, regardless of whether these two parametric modulators were put in the same GLM or in separated GLMs (*Appendix 3—Tables 9 and 10*).

The quotient of harm divided by the number of wrongdoers correlated parametrically with the activation in brain regions involved in social cognition. Specifically, the striatum linked to value computation (*Bartra et al., 2013*; *Clithero and Rangel, 2014*) and the posterior insula (pINS) associated with inequity aversion (*Gao et al., 2018*; *Hsu et al., 2008*) exhibited negative parametric responses to the quotient (*Figure 4A*; *Appendix 3—table 11*). In contrast, no brain region showed significant responses to the product of harm and responsibility. Together with the findings from computational modeling, these fMRI results suggest that individuals integrate harm and the number of wrongdoers in the form of a quotient, rather than integrating harm and responsibility in the form of a product.

## Neural basis of emotion sensitivity

To examine the neural basis of the individual harm-driven guilt sensitivity, responsibility-driven guilt sensitivity, harm-driven shame sensitivity, and responsibility-driven shame sensitivity, we correlated the coefficient estimates of harm and responsibility on guilt and shame from the linear mixed-effect regressions (*Appendix 3—table 3* and *Appendix 3—table 4*) with the whole-brain neural activity. The whole-brain analysis revealed that the parametric responses of the theory-of-mind areas (i.e. TPJ and superior temporal sulcus [STS]) to the quotient of harm divided by the number of wrongdoers were negatively correlated with responsibility-driven shame sensitivity (*Figure 4B*; *Appendix 3—table 12*). Namely, participants who more intensively converted responsibility into shame had weaker neural responses to the average harm per person in these regions. To assess whether these brain regions were specifically involved in responsibility-driven shame sensitivity, we compared the Pearson correlations between their activity and all types of emotion sensitivities. The results demonstrated the domain

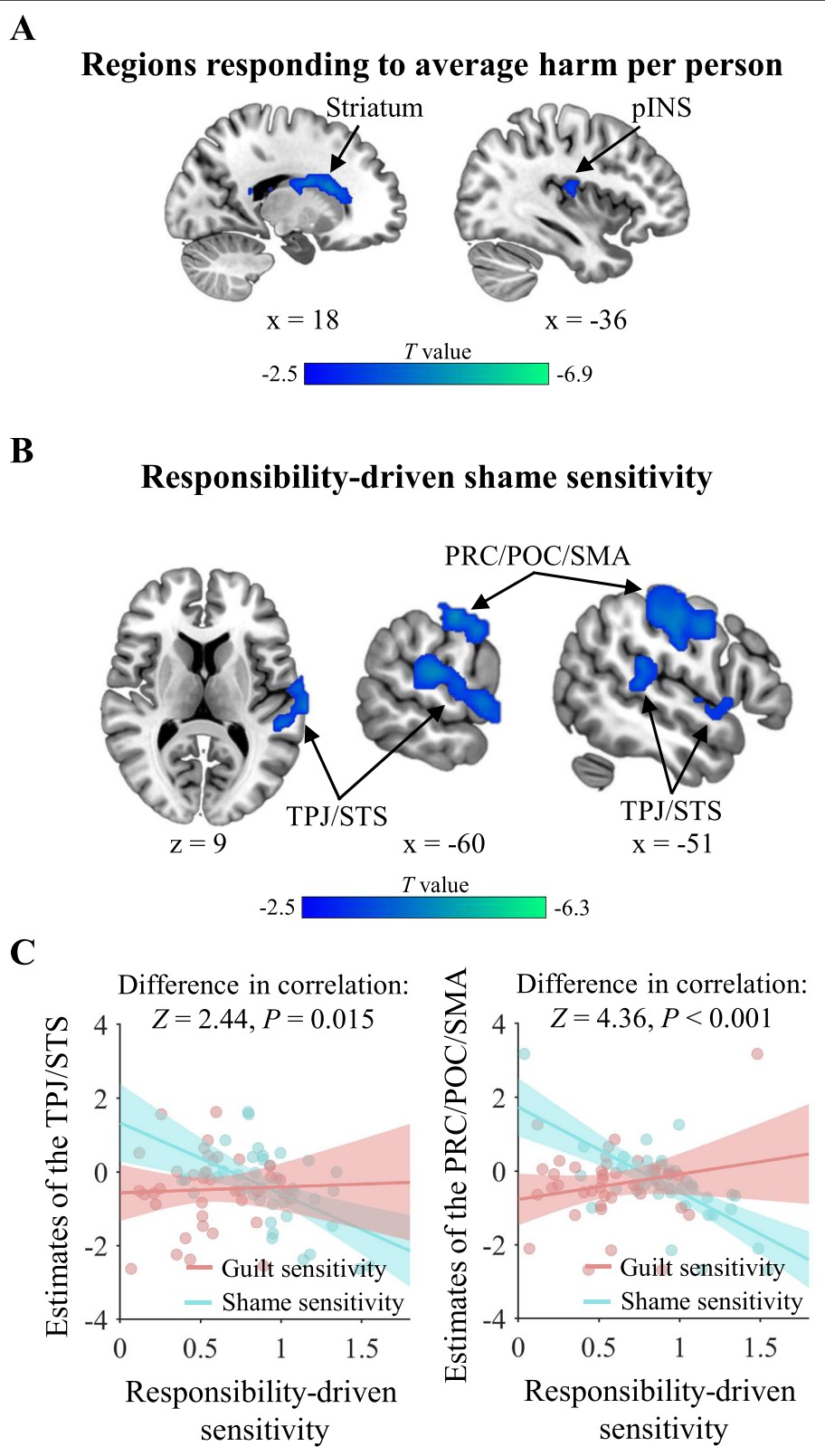

**Figure 4.** Neural representation of cognitive antecedents and neural basis of emotion sensitivity. (**A**) The quotient of harm divided by the number of wrongdoers (i.e. average harm per person) is represented by the striatum and posterior insula (pINS). (**B**) The neural responses to average harm per person in the two clusters containing temporoparietal junction (TPJ)/ superior temporal sulcus (STS) and precentral cortex (PRC)/postcentral cortex

*Figure 4 continued on next page*

*Figure 4 continued*

(POC)/ supplementary motor area (SMA) were negatively correlated with responsibility-driven shame sensitivity. (**A, B**) Negative *T* values indicate negative correlation. Whole-brain FWE-cluster correction at p<0.05 after cluster-forming-threshold at p<0.001. (**C**) The TPJ/STS and PRC/POC/SMA clusters showed a significantly stronger negative correlation with shame-driven sensitivity than with guilt-driven sensitivity. Each line represents the least squares fit with shading showing the 95% confidence interval.

specificity of these regions, by revealing that the TPJ/STS cluster had significantly stronger negative responses to responsibility-driven shame sensitivity than to responsibility-driven guilt sensitivity (*Z*=2.44, p=0.015) and harm-driven shame sensitivity (*Z*=3.38, p<0.001), and a marginally stronger negative response to harm-driven guilt sensitivity (*Z*=1.87, p=0.062; *Figure 4C*; *Appendix 3—table 13*). In addition, the sensorimotor areas (i.e. precentral cortex [PRC], postcentral cortex [POC], and supplementary motor area [SMA]) exhibited the similar activation pattern as the TPJ/STS (*Figure 4B and C*; *Appendix 3—table 12* and *Appendix 3—table 13*).

No brain response showed significant correlation with harm-driven guilt sensitivity, responsibility-driven guilt sensitivity, and harm-driven shame sensitivity.

## Neural basis of compensatory sensitivity

To examine the neural basis of the individual guilt-driven and shame-driven compensatory sensitivities, we correlated the coefficient estimates of guilt and shame on compensation from the linear mixed-effect regression (*Appendix 3—table 7*) with the whole-brain neural activity. The whole-brain analysis showed that the activation in the DMPFC/SMA was significantly correlated with both guilt-driven and shame-driven compensatory sensitivities (*Appendix 3—table 14*). Besides, the activity in the left temporal pole (TP) was significantly correlated with guilt-driven compensatory sensitivity but not with shame-driven compensatory sensitivity (*Figure 5A*), whereas activity in the bilateral inferior parietal lobe (IPL) and left LPFC clusters was significantly correlated with shame-driven compensatory sensitivity but not with guilt-driven compensatory sensitivity (*Figure 5B*). To directly assess whether these brain regions were more involved in guilt-driven or shame-driven compensatory sensitivity, we compared the Pearson correlations between brain activity and the two types of compensatory sensitivities. The results revealed that the left LPFC was more engaged in shame-driven compensatory sensitivity (*Z*=1.93, p=0.053), as its activity showed a marginally stronger positive correlation with shame-driven sensitivity than with guilt-driven sensitivity (*Figure 5C*). No significant difference was found in the Pearson correlations between the activity of the bilateral IPL and the two types of sensitivities (*Appendix 3—table 15*). For the TP, the effective sample size was too small to yield reliable results (see Materials and methods).

In addition to guilt and shame, other emotions or motivations may also contribute to the transformation of harm and responsibility into compensatory behavior. The parameter $\kappa$ from our winning computational model ought to capture the combined effects of various psychological processes on compensation, including guilt and shame. Confirming their associations, the parameter $\kappa$ significantly correlated to both guilt-driven ($\beta$=2.06, *T*(38) = 5.85, p<0.001) and shame-driven ($\beta$=2.43, *T*(38) = 2.82, p=0.008) compensatory sensitivities (*Appendix 3—table 16*). To gain a more comprehensive understanding of the neural basis of compensatory sensitivity, we correlated $\kappa$ with the whole-brain neural activity. The neural correlates of the parameter $\kappa$ largely overlapped with those associated with guilt-driven and/or shame-driven compensatory sensitivities, including the DMPFC, supplementary motor area (SMA), left TP, left LPFC (LPFC), bilateral IPL (*Figure 5A, B and D*; *Appendix 3—table 14* and *Appendix 3—table 17*). Notably, $\kappa$ is also associated with activation in the bilateral anterior insula (aINS).

## Neural correlates of trait guilt and compensation

As we observed significant correlations between trait guilt scores and compensatory behavior, we furthered our investigation by examining their neural correlates. A small-volume correction analysis revealed that repair action tendencies (i.e., a dimension of trait guilt) were significantly associated with the aMCC's responses to the quotient of harm divided by the number of wrongdoers (peak MNI coordinates: [−3, 30, 15]; cluster size: 12 voxels; $P_{FWE}$ = 0.003, small volume corrected; *Figure 6A*). The results of the aMCC, along with other clusters, retained significance after whole-brain correction

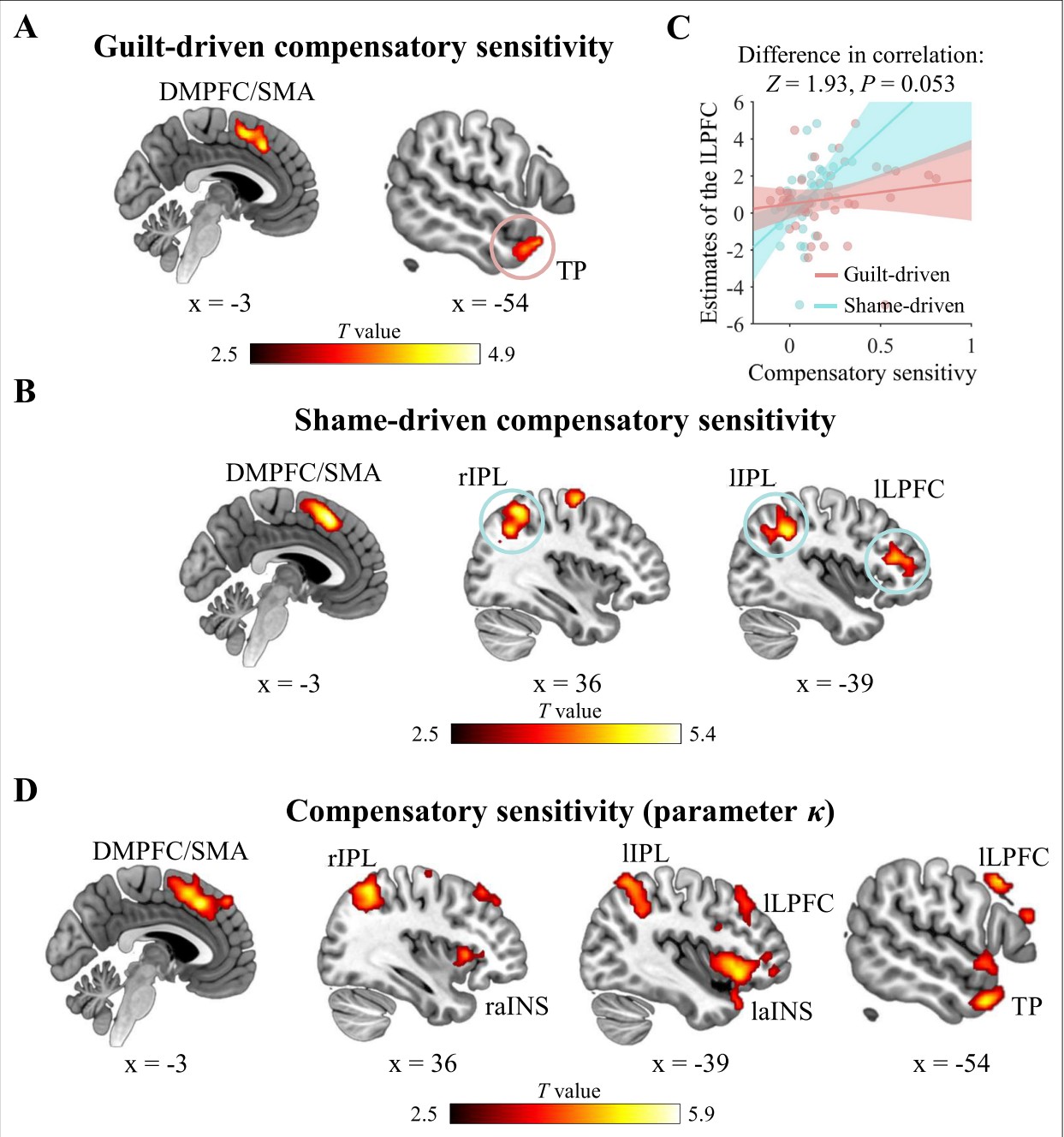

**Figure 5.** Neural basis of compensatory sensitivity. (**A**) The dorsomedial prefrontal cortex (DMPFC), supplementary motor area (SMA), and temporal pole (TP) showed significant activity associated with guilt-driven compensatory sensitivity. A red circle marked the region that showed significant activity associated with guilt-driven compensatory sensitivity but not with shame-driven compensatory sensitivity. (**B**) The DMPFC, SMA, right and left inferior parietal lobe (IPL), and left lateral prefrontal cortex (LPFC) showed significant activity associated with shame-driven compensatory sensitivity. Blue circles marked the region showed significant activity associated with shame-driven compensatory sensitivity but not with guilt-driven compensatory sensitivity. (**C**) The left LPFC showed a marginally stronger positive correlation with shame-driven sensitivity than with guilt-driven sensitivity. Each line represents the least squares fit with shading showing the 95% confidence interval. (**A, B, D**) The neural correlates of the parameter $\kappa$ largely overlapped with the regions linked to guilt-driven and shame-driven compensatory sensitivities. The parameter $\kappa$ is additionally associated with activation in the bilateral anterior insula (aINS). r, right; l, left; Whole-brain FWE-cluster correction at p<0.05 after cluster-forming-threshold at p<0.001.

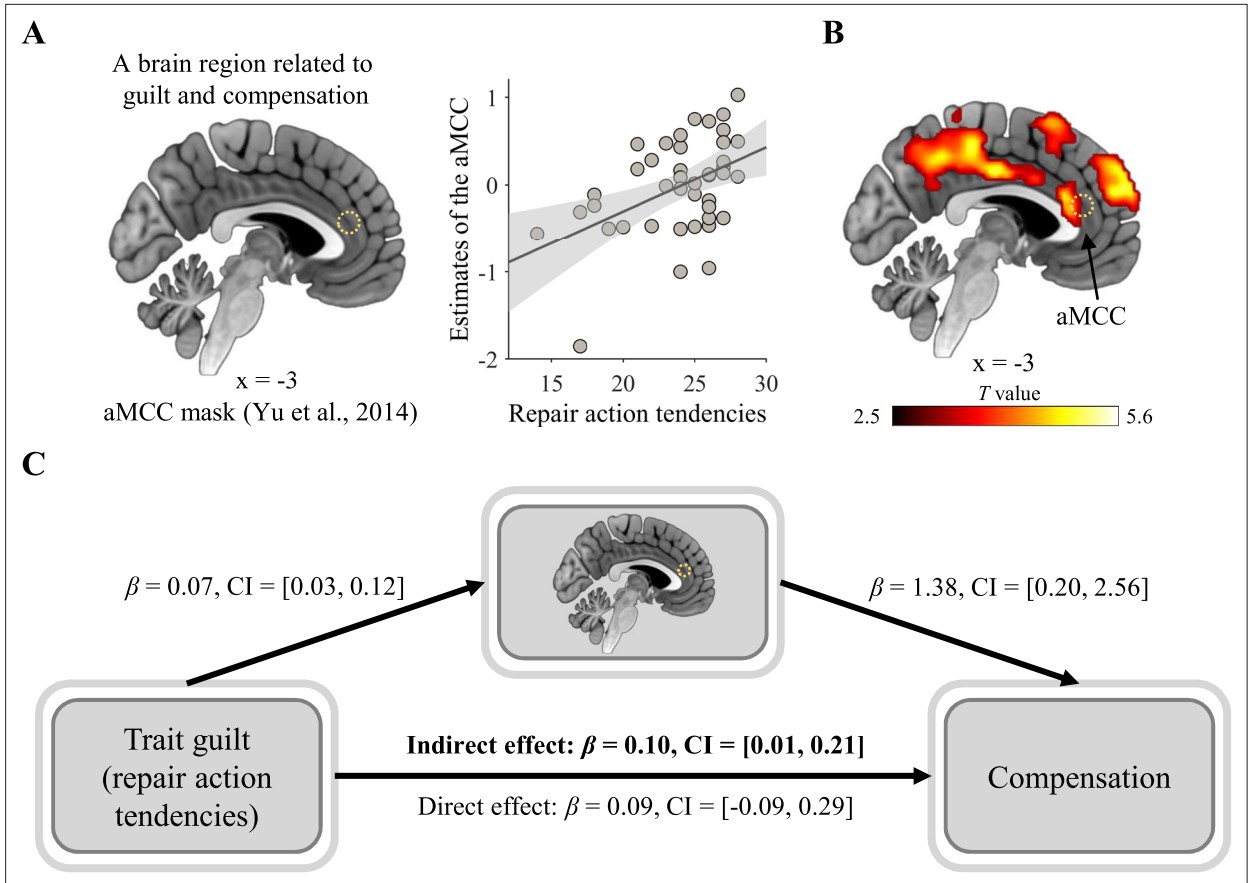

**Figure 6.** Neural correlates of trait guilt and compensation. (**A**) A small-volume correction analysis showed that participants with higher guilt trait scores (i.e., repair action tendencies) have more positive parametric responses to the quotient of harm divided by the number of wrongdoers in the anterior middle cingulate cortex (aMCC). The scatter plot is for presenting the positive correlation relationship between repair action tendencies and neural responses of the aMCC. The line represents the least squares fit with shading showing the 95% confidence interval. (**B**) The neural finding of the aMCC remained significant after whole-brain correction. Whole-brain FWE-cluster correction at p<0.05 after cluster-forming-threshold at p<0.001. (**A, B**) A circle of dots indicated the position of the aMCC mask. (**C**) The aMCC parametric responses mediated the relationship between repair action tendencies and compensation. *β*, path coefficient; CI, 95% confidence interval of *β*; bold font, significant indirect effect.

(*Figure 6B*; *Appendix 3—table 18*). Moreover, only the activity in the aMCCs mediated the relationship between repair action tendencies and compensation (indirect effect: *β*=0.10, CI = [0.01, 0.21]; *Figure 6C*).

No neural activity showed a significant correlation with negative behavior evaluations (i.e., another dimension of trait guilt), trait shame, trait gratitude, or SVO.

## Discussion

Guilt and shame have long been a focal point of research across various disciplines, including psychology, behavioral economics, neuroscience, and psychiatry (e.g. *Chang et al., 2011*; *Gao et al., 2018*; *Landers et al., 2024b*; *Mottershead et al., 2024*; *Schuster et al., 2021*; *Sznycer, 2019*; *Tangney et al., 2007*). Extensive research has investigated the psychological and neural activities associated with the experience of guilt and shame to enhance emotion regulation and improve behavioral prediction (*Bastin et al., 2016*; *Miceli and Castelfranchi, 2018*; *Michl et al., 2014*; *Piretti et al., 2023*; *Pulcu et al., 2014*; *Takahashi et al., 2004*; *Tangney and Dearing, 2003*; *Xu et al., 2022*; *Zhu et al., 2019a*). We extended this research by shifting the focus from the mere states of guilt and shame to their associations with the cognitive antecedents and behavioral consequences. Our findings advance the understanding of the psychological and neural mechanisms that underlie both the formation of guilt and shame and their subsequent transformation into compensatory behavior.

Consistent with previous studies (*Gao et al., 2021*; *Iyer et al., 2007*; *Koban et al., 2013*; *Li et al., 2020*; *Tangney, 1992*; *Yu et al., 2014*), we observed that both harm and responsibility increase individuals' feelings of guilt and shame. Of importance, for the first time, we discovered that harm exerts a stronger effect on guilt than on shame, whereas responsibility has a stronger effect on shame than on guilt. These findings provide empirical support for the proposition that guilt and shame serve distinct social functions. According to functionalist theories (e.g. *Baumeister et al., 1994*; *Gilbert, 1997*; *Sznycer, 2019*; *Sznycer et al., 2021*), guilt functions to curb the harm imposed on valued others, whereas shame functions to mitigate the cost of reputational damage to oneself. If these theories hold true, guilt should be more sensitive than shame to the information related to harm inflicted on valued others, where shame should be more sensitive than guilt to the information related to reputational damage to oneself (*Sznycer et al., 2016*). However, few studies have provided such direct evidence. An exception is the study by *Landers et al., 2024b*, which found that information related to harm inflicted on valued others (e.g., liking of a victim) and concerns about reputational damage (e.g. fear of a victim) were respectively predictive of guilt and shame. Notably, *Landers et al., 2024b* employed a vignette-based paradigm and assessed participants' guilt and shame using items that reflected the motivational tendencies characteristic of these emotions (e.g. guilt: 'I would go to him and apologize for it'; shame: 'I would feel like avoiding him for a while'). In contrast, our study utilized a laboratory-based paradigm with real-time decision-making, directly measured participants' state guilt and shame experiences, and tested new psychological factors (harm and responsibility). This approach offers greater ecological validity and provides novel evidence (*Yu et al., 2024*).

Existing findings suggest that guilt is more strongly linked to altruistic behavior than shame is (*de Hooge et al., 2008*; *de Hooge et al., 2007*; *Declerck et al., 2014*; *Gausel and Leach, 2011*; *Graton and Ric, 2017*; *Ketelaar and Tung Au, 2003*; *Tangney et al., 2007*; *Tangney and Dearing, 2003*). Nevertheless, because of methodological limitations, such as failing to compare guilt and shame directly or employing methods that may introduce confounding variables, conclusive evidence has been lacking. Overcoming these limitations, our study demonstrates that while both guilt and shame promote compensation, guilt is more effective in prompting compensatory behavior. The functionalist theories offer a framework for understanding why guilt exerts a stronger effect on compensation than shame does. Guilt corresponds to the adaptive problem of insufficiently valuing others (*Sznycer, 2019*). To address it, individuals in guilt must bring benefits to those they have harmed—typically through altruistic behavior—to correct the inequity caused by their wrongdoing. In contrast, shame is tied to reputational damage (*Sznycer, 2019*; *Sznycer et al., 2021*). Although altruistic behavior can also mitigate shame by demonstrating their social value, individuals may resort to other strategies—such as avoidance or aggression—to protect themselves from potential devaluation by others (*de Hooge et al., 2018*; *Zhu et al., 2019c*).

For a deeper understanding of social emotions, it is crucial to formally model their cognitive operations and investigate the neural underpinnings of these cognitive operations (*Yu et al., 2024*). Building on this research line, our computational modeling results reveal that individuals in guilt and shame disregard their self-interest, adopt a compensatory baseline, and mentally distribute harm across all wrongdoers. The findings not only offer a mechanistic explanation at the behavioral (algorithmic) level for guilt- and shame-driven compensatory decision-making, but also deepen the understanding of the phenomenon of responsibility diffusion by offering a formal mathematical formulation and linking it to compensatory behavior (*Darley and Latané, 1968*).

Notably, in many computational models of social decision-making, self-interest plays a crucial role (e.g. *Wu et al., 2024*). However, our computational findings suggest that participants disregarded self-interest during compensatory decision-making. A possible explanation is that the personal stakes in our task were relatively small (a maximum loss of 5 Chinese yuan), whereas the harm inflicted on the receiver was highly stigmatized (i.e. an electric shock). Under conditions where the harm is highly salient and the cost of compensation is low, participants may be inclined to disregard self-interest and focus solely on making appropriate compensation.

At the neural level, our findings demonstrate the involvement of the posterior insula and striatum in representing the cognitive antecedents of guilt and shame. Specifically, the activation in these brain regions decreased as the quotient of harm divided by the number of wrongdoers increased. Harm inflicted on the victim, particularly the portion for which a wrongdoer is responsible, creates a sense of inequity between them. Beyond its well-established role in interoceptive awareness (*Craig, 2009*;

*Craig, 2011*), the pINS has been implicated in processing economic inequity in allocation tasks (*Gao et al., 2018*; *Hsu et al., 2008*). For instance, *Hsu et al., 2008* reported that pINS activation negatively correlates with the degree of inequity, suggesting that greater inequity elicits lower pINS activation. Our results extend this role of the pINS beyond economic inequity to encompass harm inequity. Given that the striatum is implicated in value computation (*Bartra et al., 2013*; *Crockett et al., 2017*; *Rilling et al., 2008*), we believe that its activity reflects individuals' perception of the loss (i.e. harm) inflicted on the victim. In contrast, no brain region had significant responses to the product of harm and responsibility. Thus, the fMRI and computational modeling findings offer convergent evidence indicating that individuals are more likely to integrate these cognitive antecedents in a form of quotient.

In addition, no brain region exhibited significant responses to harm. Only the sensorimotor areas showed significant responses to the number of wrongdoers (i.e. the complement of responsibility, 5 − responsibility level). Although the fMRI findings revealed that no brain region associated with social cognition showed significant responses to harm or responsibility, this does not suggest that the human brain encodes only a unified measure integrating harm and responsibility and does not process them as separate entities. Using more fine-grained techniques, such as intracranial electrophysiological recordings, it may still be possible to observe independent neural representations of harm and responsibility.

As to emotion sensitivity, our findings show that individuals who tend to convert responsibility into shame exhibit reduced activation in brain regions associated with other-oriented theory-of-mind processing, specifically the TPJ and STS. The TPJ and STS have been implicated in inferring others' mental states (*Schurz et al., 2014*). Lower activation in these regions indicates that individuals with higher responsibility-driven shame sensitivity may be less engaged in considering the victim's experiences and thoughts. This aligns with existing research on shame, which, compared to guilt, is associated with less concerns on one's actions on others (*Tangney and Dearing, 2003*) and weaker activation in the TPJ (*Zhu et al., 2019a*).

Regarding compensatory sensitivity, our results show that both individuals with higher guilt-driven and shame-driven compensatory sensitivity have stronger activation in the DMPFC. This region is central to both theory-of-mind processing (*Schurz et al., 2014*) and self-referential processing (*Northoff et al., 2006*), playing a crucial role in combining others' thoughts and feelings with one's own (*D'Argembeau et al., 2007*; *Saxe et al., 2006*). *Zhu et al., 2019a* have identified the DMPFC's involvement in the experience of both guilt and shame. Our findings here further highlight its role in translating these emotions into compensatory behavior.

We found that the TP's activity is positively related to individuals' guilt-driven compensatory sensitivity. This region is considered a core part of the theory-of-mind network (*Frith and Frith, 2003*). Numerous studies suggest that the activation in this region reflects retrieval of both general conceptual knowledge (*Lambon Ralph and Patterson, 2008*) and social conceptual knowledge (e.g. social rules; *Ross and Olson, 2010*; *Sugiura et al., 2006*; *Tsukiura et al., 2010*; *Zahn et al., 2007*). The retrieval of such information likely facilitates understanding others' thoughts and empathizing with their suffering (*Olson et al., 2007*; *Schurz et al., 2014*). Empathy, in turn, has been widely established as a significant driver of altruistic behavior, including compensatory behavior (*Ding and Lu, 2016* ; *Eisenberg and Miller, 1987*). Our findings confirm the involvement of the TP in translating guilt into compensation.

The IPL's activity has a positive correlation with individuals' shame-driven compensatory sensitivity. This region is associated with various non-social and social cognitive functions, including number processing (*Pinel et al., 2004*; *Sandrini et al., 2004*), salience processing (*Kahnt et al., 2014*), and theory-of-mind processing (*Igelström and Graziano, 2017*; *Tusche et al., 2016*). Two recent studies provided direct evidence showing that the IPL plays a role in encoding others' benefits during altruistic decision making (*Hu et al., 2017*; *Hu et al., 2021*). Our findings about IPL can be explained by its involvement in generating other-regarding motives (*Hu et al., 2021*) that facilitate the conversion from shame to compensation.

We did not find a significant difference in the correlations between TP activity and guilt-driven versus shame-driven compensatory sensitivities. Similarly, no significant difference was observed in the correlations between IPL activity and shame-driven versus guilt-driven compensatory sensitivities. These findings suggest that neither of these regions plays a domain-specific role in compensation driven by guilt or shame.

In contrast, LPFC activity exhibited a stronger correlation with shame-driven compensatory sensitivity than with guilt-driven compensatory sensitivity, indicating a domain-specific role of the LPFC in shame-related compensation. The LPFC is implicated in cognitive control (*Koechlin et al., 2003*) and the optimization of social decision-making (*Buckholtz and Marois, 2012*; *Feng et al., 2015*). Some brain stimulation studies have demonstrated that disrupting LPFC activity impairs individuals' ability to inhibit selfish or aggressive impulses, which can incur social devaluation and punishment from others (*Knoch et al., 2006*; *Knoch et al., 2009*; *Riva et al., 2015*). Further research has extended these findings by emphasizing the LPFC's role in strategic social behavior (*Ruff et al., 2013*; *Strang et al., 2015*). For instance, *Ruff et al., 2013* found that when individuals face the possibility of being punished for selfish behavior, enhancing LPFC activity suppresses selfish impulses and promotes altruistic behavior. However, in the absence of punishment risk, enhancing LPFC activity instead reduces altruistic behavior and promotes self-interest. Considering that guilt is typically alleviated through altruistic behavior (*Tang et al., 2019*), whereas coping strategies for shame are more varied—ranging from altruistic behavior to aggression and avoidance (*Sznycer, 2019*; *Sznycer et al., 2016*)—shame appears to be more closely linked to strategic thinking than guilt. This explains why LPFC activity, which is associated with strategic behavior, is more strongly related to shame-driven compensatory sensitivity than to guilt-driven compensatory sensitivity.

The neural correlates of the parameter $\kappa$ largely overlapped with those linked to compensatory sensitivities driven by guilt and shame. Intriguingly, beyond that, $\kappa$ also showed a strong association with aINS activity. Insula, known as a key node in the salience network (*Uddin, 2015*), engages in during experiencing various negative emotions, including sadness (*Wagner et al., 2011*), disgust (*Craig, 2009*), guilt (*Yu et al., 2014*), shame (*Piretti et al., 2023*; *Zhu et al., 2019a*), and indebtedness (*Gao et al., 2024*). Social neuroscience research highlights the critical role of the aINS in processing norm violations and guiding behavior accordingly (*Bellucci et al., 2018*; *Zinchenko and Arsalidou, 2018*). For example, *Chang et al., 2011* found that aINS serves to mitigate anticipated negative feelings triggered by norm violations by facilitating individuals' reciprocity toward their partners' investments in a trust game, thereby maintaining adherence to the norm of reciprocity. Consistently, numerous studies on the ultimatum game reveal aINS's involvement in rejecting unfair offers and upholding the norm of fairness (*Feng et al., 2015*; *Gabay et al., 2014*). In the same line, the findings on the involvement of aINS in social conformity also manifest its role in monitoring norm violations and reinforcing adherence to social norms (*Berns et al., 2010*; *Klucharev et al., 2009*). Given the involvement of the aINS in various social-affective processes, our findings suggest that the motivation to uphold social norms might directly shape individuals' compensatory behavior or indirectly influence it through emotions beyond guilt and shame, with aINS activity playing a pivotal role in this process.

Interestingly, we found that the sensorimotor areas were associated with the representation of a shame-related cognitive antecedent (i.e. responsibility) and emotional sensitivity. Our findings align with the result from a fMRI meta-analysis, which identified the involvement of sensorimotor regions in processing shame (*Piretti et al., 2023*). It has been suggested that sensorimotor activation may reflect typical shame-related action tendencies, such as reduced social presence, speech, and movement (*Piretti et al., 2023*). However, in our study, participants were required to remain completely still throughout the experiment to maintain MRI data quality and were continuously monitored, eliminating the possibility of physical withdrawal. Therefore, the observed sensorimotor activation may reflect motor preparation for subsequent button-press responses associated with compensation rather than a general tendency toward shame-related avoidance. Future studies that permit participants to engage in actual avoidance behaviors could further clarify the role of sensorimotor areas in shame processing.

In line with previous research (*Cohen et al., 2011*), our findings reveal that both dimensions of trait guilt were significantly associated with compensatory behavior, whereas neither dimension of trait shame exhibited such an association. Furthermore, we found neural responses in the aMCC mediated the relationship between repair action tendencies (one dimension of trait guilt) and compensation. A substantial body of research has revealed that guilt processing consistently activates the aMCC (see a meta-analysis, *Gifuni et al., 2017*). It is suggested that, in the context of guilt, the aMCC plays a role in detecting the conflict between social norms and actual behavior and signaling this conflict via generating negative emotions (*Bastin et al., 2016*; *Gifuni et al., 2017*). In addition, *Yu et al., 2014* linked the aMCC activity with compensatory behavior. Accordingly, our fMRI findings

suggest that individuals with stronger tendency to engage in compensation across various moral violation scenarios (indicated by their repair action tendencies) are more sensitive to the severity of the violation and therefore engage in greater compensatory behavior. However, the neural correlates of negative behavior evaluations (another dimension of trait guilt) were absent. The reasons underlying the non-significant neural finding may be multifaceted. One possibility is that negative behavior evaluations influence neural responses indirectly through intermediate processes not captured in our study (e.g. specific motivational states).

Although previous research has found that trait gratitude and SVO are significantly associated with altruistic behavior in contexts such as donation (*Van Lange et al., 2007*; *Yost-Dubrow and Dunham, 2018*) and reciprocity (*Ma et al., 2017*; *Yost-Dubrow and Dunham, 2018*), their associations with compensatory decisions in the present study were not significant. This suggests that the effects of trait gratitude and SVO on altruistic behavior are context-dependent and may not predict all forms of altruistic behavior.

This research has several limitations. First, post-task assessments of guilt and shame, unlike in-task assessments, rely on memory and may thus be less precise, although in-task assessments could have heightened participants' awareness of these emotions and biased their decisions. Second, our measures of guilt and shame depend on participants' conceptual understanding of the two emotions. While this is common practice in studies with adult participants (*Michl et al., 2014*; *Wagner et al., 2011*; *Zhu et al., 2019a*), it may be less appropriate for research involving children. Third, although we aimed to construct a conceptually comprehensive computational model space informed by prior research and our own understanding, it does not encompass all plausible models. Future research is encouraged to explore additional possibilities. Fourth, fMRI cannot establish causality. Future studies using brain stimulation techniques (e.g. transcranial magnetic stimulation) are needed to clarify the causal role of brain regions in guilt-driven and shame-driven altruistic behavior. Fifth, we did not explicitly measure emotions similar to guilt and shame (e.g. indebtedness), which would have been helpful for understanding their distinct contributions. Sixth, marginally significant results should be viewed cautiously and warrant further examination in future studies with larger sample sizes.

Our study makes several innovative contributions. First, we developed a novel paradigm that effectively elicits guilt and shame at comparable intensities, enabling researchers to systematically explore the associations among guilt, shame, their cognitive antecedents, and behavioral consequences. Future research could combine this paradigm with other cognitive neuroscience methods, such as electroencephalography (EEG) or magnetoencephalography (MEG), and adapt it to investigate additional behaviors linked to guilt and shame, including donation (*Xu, 2022*), avoidance (*Shen et al., 2023*), and aggression (*Velotti et al., 2014*). Second, our behavioral findings provide high-quality empirical evidence for functionalist theory, aligning with the contemporary trend of comprehending emotions through their adaptive functions (*Landers et al., 2024b*; *Sznycer et al., 2021*). Third, our computational and neural findings offer a clear delineation of the neurocognitive mechanisms underlying guilt and shame. Building on knowledge that harm and responsibility are related to guilt and shame, our results further reveal how these cognitive antecedents are integrated. While previous studies have broadly identified brain regions associated with guilt and shame processing as a whole (*Bastin et al., 2016*; *Gifuni et al., 2017*; *Piretti et al., 2023*), our study advances this understanding by breaking down guilt and shame processing into distinct processes and precisely mapping the neural correlates of each process.

Our study has potential practical implications. The behavioral findings may help counselors understand how cognitive interventions targeting perceptions of harm and responsibility could influence experiences of guilt and shame. The neural findings highlight specific brain regions (e.g. TPJ) as potential intervention targets for regulating these emotions. Given the close links between guilt, shame, and various psychological disorders (e.g. *Kim et al., 2011*; *Lee et al., 2001*; *Schuster et al., 2021*), strategies to regulate these emotions may contribute to symptom alleviation. Nevertheless, because this study was conducted with healthy adults, caution is warranted when considering applications to other populations.

In conclusion, our findings support the functionalist theory by demonstrating distinct effects of harm and responsibility on guilt and shame, as well as differences in the efficiency with which guilt and shame translate into compensatory behaviors. Notably, harm and responsibility are integrated in a manner consistent with responsibility diffusion prior to influencing guilt-driven and shame-driven

compensation. Furthermore, the distinct stages involved in guilt and shame processing correspond to activities in specific neural regions related to value computation, salience processing, theory-of-mind processing, self-referential processing, and cognitive control. By simultaneously providing computational, algorithmic, and neural accounts of guilt and shame (*Yu et al., 2024*), our study advances the holistic understanding of these emotions, which provides insights into how guilt and shame can be regulated and informs the treatment of guilt- and shame-related mental disorders.

# Materials and methods

## Key resources table

| Reagent type (species) or resource | Designation | Source or reference | Identifiers | Additional information |
|---|---|---|---|---|
| Software, algorithm | MATLAB | MATLAB | RRID:SCR_001622 | |
| Software, algorithm | SPM 12 | SPM 12 | RRID:SCR_007037 | |
| Software, algorithm | R Project for Statistical Computing | R Project for Statistical Computing | RRID:SCR_001905 | |

## Participants

We recruited 49 college students. All participants had normal or corrected-to-normal vision and reported no history of psychological or neurological disorders. They provided written consent and gained monetary payment for their participation. The experimental protocol (protocol number: ICBIR_A_0071_010) was approved by the local research ethics committee at the State Key Laboratory of Cognitive Neuroscience and Learning, Beijing Normal University and was conducted in accordance with the Declaration of Helsinki.

Data from four participants weren't correctly recorded due to a machine malfunction. One participant dropped out of the experiment for personal reasons, one missed more than half of the experimental trials, and one fell asleep in the fMRI scanner. These seven participants were removed from all analyses. In addition, one participant who didn't compensate the receiver in any trial (i.e. all compensation decisions were 0) was excluded from the computational modeling because the model parameters couldn't be reliably estimated under such a condition (*Zhong et al., 2016*). Thus, unless otherwise specified, analyses unrelated to computational modeling were conducted with 42 participants (18 females, 24 males; age: $M \pm SD$ = 21.71±2.30 years), while analyses related to computational modeling were conducted with 41 participants (18 females, 23 males; age: $M \pm SD$ = 21.71±2.33 years).

## Experimental procedures

The experiment comprised two phases (*Figure 1A*). In the first phase, participants completed personality questionnaires online, including the Guilt and Shame Proneness scale (GASP; *Cohen et al., 2011*), Gratitude Questionnaire–6 scale (GQ-6; *Mccullough et al., 2002*), and Social Value Orientation Slider Measure (SVO Slider Measure; *Murphy et al., 2011*). The 16-item GASP measures trait guilt and trait shame. Participants read various scenarios and indicated the likelihood of specific responses on a 7-point Likert scale (1=very unlikely, 7=very likely). The scale includes two guilt subscales—negative behavior evaluations (e.g. 'What is the likelihood that you would feel terrible about the lies you told?') and repair action tendencies (e.g. 'What is the likelihood that you would try to act more considerately toward your friends?')—and two shame subscales—negative self-evaluations (e.g. 'What is the likelihood that this would make you feel like a bad person?') and withdrawal action tendencies (e.g. 'What is the likelihood that you would avoid the guests until they leave?'). The six-item GQ-6 is a widely used measure of trait gratitude. Each item (e.g. 'I am grateful to a wide variety of people') is answered on a 7-point Likert scale (1=strongly disagree, 7=strongly agree). The six-item SVO Slider Measure assesses social value orientation, defined as the tendency to prioritize and balance outcomes for oneself and others in interdependent situations (*Murphy and Ackermann, 2014*). Participants chose among several self-other payoff combinations. Based on their choices, an SVO score was calculated. Larger SVO scores indicate more prosocial orientation. The participants filled out the questionnaires online at least 1 day before the second phase.

We measured trait gratitude and SVO for exploratory purposes. Previous research has shown that both are linked to altruistic behavior, particularly in donation contexts (*Van Lange et al., 2007*; *Yost-Dubrow and Dunham, 2018*) and reciprocity contexts (*Ma et al., 2017*; *Yost-Dubrow and Dunham, 2018*). Here, we explored whether they also exert significant effects in a compensatory context.

In the second phase, each participant arrived at our laboratory individually. Upon arrival, the participant was introduced in person to three co-players. These co-players, however, were confederates, and they were purported to interact with the participant later via an internal network. At least one confederate shared the participant's gender, and at least one was of a different gender. The participant was then led to a separate room and completed a series of tasks.

### Pain calibration

First, the participant underwent an individual pain calibration procedure with an SXC-4A multichannel electrical stimulator. After introducing the calibration process and precautions, we cleaned the participant's left forearm and placed two disposable electrodes on the back of their left wrist. The initial electric shock was set as 12 repeated square waveform electrical stimulation pulses (current intensity of each pulse: 0.2 mA; duration of each pulse: 0.5ms; interval between pulses: 10ms; *Yu et al., 2018*). We gradually increased or decreased the current intensity of each pulse in small increments (0.5 mA) with a 3:1 ratio (*Crockett et al., 2014*). The participant rated their subjective experience after each shock on an 11-point Likert scale (0=no sensation, 10=maximum tolerable pain). The calibration procedure continued until the participant reported a rating of 8. The current intensity value corresponding to this self-reported pain rating of 8 was recorded and used in the following steps. We then adjusted the number of pulses and applied electric shocks involving 2, 4, 8, and 12 repeated pulses to the participant (*Yu et al., 2018*). The participant rated the subjective pain intensity for each of the four shock levels. They were informed that these four pain levels would be used in the upcoming task. Providing the participant with the opportunity to experience the electric shocks beforehand enhanced the experiment's authenticity.

### Interpersonal game (with fMRI scanning)

After the pain calibration, the participant played a novel interpersonal game while their blood-oxygen-level-dependent (BOLD) signals were measured using fMRI. The development of this game was inspired by several previous studies on guilt and shame (*Li et al., 2020*; *Yu et al., 2014*; *Zhu et al., 2019a*; *Zhu et al., 2019b*). In the game, there were five players: four deciders and one receiver. At the beginning of each trial, a picture of dots was presented on each decider's computer screen (dots period, 1.5 s; *Figure 1B*). Each decider indicated whether the number of dots was more than or less than 20 based on their own estimation by pressing a corresponding button (dots estimation period, <2.5 s) and was unaware of the estimations made by other deciders. Unbeknownst to the participant, the picture always contained 20 dots and the positions of the dots varied across trials. The four deciders wouldn't receive any electric shock regardless of the correctness of their estimations. However, the receiver would receive an electric shock if any decider made an incorrect estimation. The electric shock had four levels corresponding to the pain levels in the pain calibration procedure (pain levels 1–4). It was emphasized that all electric shocks were within safe limits and we had measured the pain sensitivity of all players to ensure that the deciders could experience the receiver's pain sensation caused by different levels of electric shocks. The pain level of each trial was randomly determined. After making their estimations, the deciders saw the pain level of the current trial (cue period, 1–3 s), which was indicated by the number of white squares in the four white frames next to the electric shock symbol (e.g. three white squares indicating pain level 3). Afterward, the deciders viewed the outcome (outcome evaluation period, 4.5 s). The first row displayed whether the receiver was subjected to an electric shock and its pain level. If the receiver wouldn't receive any electric shock (i.e. all deciders made correct estimations), the white squares in the white frames disappeared. If the receiver would receive an electric shock, the white squares turned into black and filled the white frames. The second row showed whether the current decider made a correct estimation. A tick appeared in the black frame for a correct estimation, while a cross appeared for an incorrect estimation. The third row revealed how many other deciders, excluding the current decider, made incorrect estimations. It was represented by the number of black squares in the three white frames next to the small cross symbol. At the end of each trial, each decider was endowed 10 tokens (1 token = 0.5 Chinese yuan) and could

freely distribute the tokens between themselves and the receiver (minimum unit: 1 token) by moving a black block using a response box (decision-making period, <6 s). The initial position of the block was randomized across trials, which helped minimize stable anchoring effects across trials. When the decider finalized their decision by pressing a button, the black block turned red as feedback for their response (decision feedback period, 6 s - response time). Each decider made their distribution independently and didn't know how many tokens the other deciders gave to the receiver. The receiver could receive a maximum of 40 tokens in a trial. Five trials would be randomly chosen and actualized (i.e. implementing the electric shocks and monetary rewards) at the end of the interpersonal game.

The participant and three confederates played as the deciders. We told the participant a cover story that the receiver was played by another college student who was not present in the laboratory at the time. They had visited the laboratory, undergone pain calibration, spent time understanding the game rules, and agreed to participate as the receiver. The receiver would visit the laboratory again to see the behaviors of all the deciders and face the consequences, including electric shocks and monetary rewards. In fact, the receiver did not actually exist. Not having participants meet the receiver helped prevent excessive guilt and shame that might produce a ceiling effect, while also eliminating the need to recruit an additional confederate.

The outcomes of the interpersonal game were predetermined. The experimental trials were those in which the participant as a decider was informed that they made an incorrect estimation (i.e. the self-error condition). We had a 4 (harm: levels 1–4)×4 (responsibility: levels 1–4) within-subjects design. To create different levels of harm and responsibility, we manipulated the pain levels of the electric shock [harm level 1 (pain level 1), harm level 2 (pain level 2), harm level 3 (pain level 3), and harm level 4 (pain level 4)] and the number of other deciders who also made incorrect estimates: responsibility level 1 (3 deciders), responsibility level 2 (2 deciders), responsibility level 3 (1 decider), and responsibility level 4 (0 deciders), respectively. There were 16 possible combinations of harm levels and responsibility levels. We repeated these combinations 4 times and thus built 64 experimental trials. Unless otherwise specified, only data from these experimental trials were included in our analyses. To enhance the authenticity of the experiment, we also involved 48 filler trials, in which the participant was informed that they made a correct estimation (*Appendix 3—table 19*). Thus, the interpersonal game contained a total of 112 trials. We divided the trials into 4 sessions. Each session involved 16 experimental trials and 12 filler trials. The trials of a specific session were presented in a pseudo-random order. The 4 sessions were assigned to 4 fMRI scanning runs in a balanced manner across participants.

Participants did not proceed to the interpersonal game until they had fully understood the experimental rules and passed a comprehension test.

## Post-game survey (guilt and shame feelings recalling)

After the interpersonal game, the outcomes of the experimental trials were re-presented in a random order. All the participants were required to recall and rate their feelings of guilt and shame (emotion ratings) when they saw those outcomes during the game (0=not at all, 10=very strong; 11-point Likert scale). They gave a rating of guilt and a rating of shame for each harm-responsibility combination (i.e. each participant provided 16 guilt ratings and 16 shame ratings).

## Post-game survey (responsibility manipulation check)

The participants were also asked to rate their perceived responsibility for causing the receiver's harm (1) when they were the only decider who made an incorrect estimation; (2) when they were one of the two deciders who made incorrect estimations; (3) when they were one of the three deciders who made incorrect estimations; (4) when they were one of the four deciders who made incorrect estimations; and (5) when they made a correct estimation, but the other deciders made wrong estimations (1=no responsibility, 9=high responsibility, 9-point Likert scale).

## Post-game survey (additional measures and procedures)

Participants rated the perceived difficulty of the dot estimation task (1=not at all, 4=very much, 4-point Likert scale; M=3.07, SD = 0.46) (*Yu et al., 2014*) and the extent of variation in task difficulty across trials (1=not at all, 4=very much, 4-point Likert scale; M=2.43, SD = 0.59). Additionally, they reported to what extent they wanted to make correct estimations in the dot estimation task (1=not at all, 9=very much, 9-point Likert scale; M=7.45, SD = 1.56) and to what extent they wanted to help the

receiver to avoid electric shocks (1=not at all, 9=very much, 9-point Likert scale; $M$=7.40, SD = 1.99). The results suggest that the participants found the task moderately challenging with some variation in difficulty across trials, while remaining highly motivated to perform accurately and positively impact the receiver, reflecting their active engagement in the task.

To examine whether the participants had any suspicion of the experiment, we conducted an interview at the very end of the experiment. No participant expressed doubts about the authenticity of the experiment.

## Magnetic resonance imaging data acquisition

Whole-brain imaging data were acquired using a 3-Tesla Prisma magnetic resonance scanner at the State Key Laboratory of Cognitive Neuroscience and Learning & IDG/McGovern Institute for Brain Research, Beijing Normal University. During the interpersonal game, T2-weighted echo-planar functional images were acquired (TR/TE: 2000/30ms; FoV: [224 mm (RL) 224 mm (AP) 143 mm (FH)]; flip angle: 90°; voxel size: 2×2 × 2 mm; number of slices: 62; slice thickness: 2 mm; slice gap: 0.3 mm). Additionally, a T1-weighted 3-dimensional magnetization-prepared rapid acquisition gradient echo structural image was collected for image registration (TR/TE: 2530/2.98ms; FoV: [192 mm (RL) 224 mm (AP) 256 mm (FH)]; flip angle: 7°; voxel size: 0.5×0.5 × 1 mm; number of slices: 192; slice thickness: 1 mm; slice gap: 0.5 mm) and gradient-echo field map images were acquired for correcting deformations (TR/TE1/TE2: 620/4.92/7.38ms; FoV: [224 mm (RL) 224 mm (AP) 143 mm (FH)]; flip angle: 60°; voxel size: 2×2 × 2 mm; number of slices: 62; slice thickness: 2 mm; slice gap: 0.3 mm).

## Behavioral analyses

### Harm and responsibility manipulation checks

Painful feelings caused by electric shocks can be regarded as a form of harm (e.g. *Crockett et al., 2017*). To examine whether the electric shocks induced distinguishable perceptions of pain, we compared participants' pain ratings pairwise across four levels of harm by conducting repeated measures ANOVA. To examine whether the number of other deciders who made incorrect estimates with the participant caused distinguishable perceptions of responsibility, we compared participants' responsibility ratings pairwise across four levels of responsibility by conducting repeated measures ANOVA. Additionally, to confirm that participants' perceptions of responsibility were influenced by their correctness of estimates, we compared participants' responsibility ratings between conditions where they made incorrect and correct estimates using repeated measures ANOVA.

### The effects of harm and responsibility on guilt and shame

To test the influence of harm and responsibility on guilt and shame, we performed two separate linear mixed-effect regressions. In the first regression, we regressed participants' emotion ratings onto harm levels, emotion types (guilt vs. shame), and their interaction. In the second regression, we regressed participants' emotion ratings onto responsibility levels, emotion types (guilt vs. shame), and their interaction. The significance of the interaction effects in these two models allowed us to evaluate whether harm and responsibility had distinct influence on guilt and shame, which is a key question we are concerned with. We also conducted simple slope analyses for each emotion type to test whether harm and responsibility had significant effects on guilt and shame, respectively. As a complementary analysis, we conducted another linear mixed-effect regression, in which participants' emotion ratings were regressed onto harm levels, responsibility levels, emotion types (guilt vs. shame), and all possible interactions. We used it to test whether the interaction between harm and emotion type, as well as the interaction between responsibility and emotion type, remained significant when controlling for the effects of other regressors. For all the regressions above, participant-specific random constants and random slopes for each fixed effect were involved as random effects.

### The effects of guilt and shame on compensation

To examine whether and how guilt and shame affect compensation, we built four linear mixed-effect regression models. In all models, the dependent variable remained the same: the amount of compensation (i.e. the number of tokens that participants distributed to the receiver), while the fixed-effect regressors varied. Model I included guilt rating as the sole regressor. Model II included shame rating

as the sole regressor. Model III included both guilt and shame ratings. Model IV included guilt rating, shame rating, and their interaction. In all the regression models, participant-specific random constants and random slopes for each fixed effect were involved as random effects. The model with the lowest Bayesian information criterion (BIC) value was selected as the best model. Model III won, which suggests guilt and shame each uniquely contribute to compensatory behavior. To further understand the effects of guilt and shame, we used Wald chi-square tests to directly compare their influence (i.e. regression coefficients) on compensation based on the best model (i.e. Model III). Besides, to test whether the difference in the effects of guilt and shame on compensation was due to the difference in their intensity, we compare participants' average guilt and shame ratings across trials using repeated measures ANOVA.

## Correlation analyses

Although the current study focused on state guilt and shame, we were aware of prior findings suggesting that trait guilt is associated with moral behavior, whereas trait shame seems not (**Cohen et al., 2011**; **Tangney and Dearing, 2003**). To replicate these findings, we calculated Pearson correlation coefficients to examine the relationship between the amount of compensation and trait guilt and trait shame. Additionally, we conducted exploratory analyses by calculating Pearson correlation coefficients between the amount of compensation and trait gratitude and social value orientation.

## Computational modeling

### Model development

After validating the associations among harm, responsibility, guilt, shame, and compensation through separate linear mixed-effect regressions, we sought to advance mechanistic understanding of guilt-driven and shame-driven compensatory decision-making through computational modeling. Drawing from the phenomenon of responsibility diffusion (**Darley and Latané, 1968**), we hypothesized that individuals perceive responsibility as being diluted when multiple wrongdoers are involved. This implies a cognitive mechanism wherein individuals mentally distribute harm across all wrongdoers, thereby reducing their altruistic behavior (compensation). Based on this Diffusion Hypothesis, we constructed Model family 1, where harm and the number of wrongdoers are integrated in the form of a quotient. All models within this family share this psychological mechanism, differing only in whether self-interest was accounted for (e.g. **Wu et al., 2024**) and whether a compensatory baseline was incorporated (e.g. **Feng et al., 2023**), both of which are common psychological components involved in social decision-making.

The models of Model family 1 are described as below:

Model family 1 (Diffusion Hypothesis):
Model 1.1 (self-interest considered and compensatory baseline exists):

$$U\left(D\right) = \theta * \left(10 - D\right) - \left(1 - \theta\right) * |\kappa * \frac{H}{W} + \eta - D|$$

where $U\left(D\right)$ is the utility for a compensatory decision. $D$ is the number of tokens distributed to the receiver by participants, and $10 - D$ is the number of tokens left for participants themselves (i.e. self-interest). $H$ and $W$ are the level of harm and the number of wrongdoers, respectively. $W$ is the complement transformation of the level of responsibility ($W$=5 – the level of responsibility). $\frac{H}{W}$ denotes the average harm for which participants are responsible, suggesting participants integrated harm and the number of wrongdoers in the form of a quotient. The parameter $\kappa$ represents compensatory sensitivity. Larger $\kappa$ values indicate that participants are willing to allocate more tokens to compensate for the harm they are responsible for. The parameter $\eta$ represents compensatory baseline. Larger $\eta$ values indicate participants are inclined to allocate more tokens to the receivers for compensation, irrespective of the level of harm and responsibility. $\kappa * \frac{H}{W} + \eta$ is the total number of tokens that participants believe should be distributed to the receiver as compensation. $-|\kappa * \frac{H}{W} + \eta - D|$ means that participants are averse to providing the receiver less compensation (undercompensation) or more compensation (overcompensation) they believe the receiver deserves (i.e., improper compensation aversion). The parameter $\theta$ represents greed. Larger $\theta$ values indicate participants put more weight on self-interest (relative to improper compensation aversion). Model 1.1 is the full model in model family

1, while the following Model 1.2, Model 1.3, and Model 1.4 are simplified versions, each omitting specific components.

Model 1.2 (self-interest considered and compensatory baseline doesn't exist):

$$U\left(D\right) = \theta * \left(10 - D\right) - \left(1 - \theta\right) * |\kappa * \frac{H}{W} - D|$$

Model 1.3 (self-interest not considered and compensatory baseline exists):

$$U\left(D\right) = -|\kappa * \frac{H}{W} + \eta - D|$$

Model 1.4 (self-interest not considered and compensatory baseline doesn't exist):

$$U\left(D\right) = -|\kappa * \frac{H}{W} - D|$$

It is also plausible that individuals might perceive their responsibility as intensifying the impact of harm (**Moore, 2015**). Under this mechanism, harm and responsibility interact in a multiplicative manner, resulting in an amplified sense of moral obligation and increased compensatory behaviors. Based on this Amplification Hypothesis, we built Model family 2. The models in the Model family 2 were the same as the Model family 1, except that the quotient was replaced by the product of harm and responsibility. The models of Model family 2 are described as below:

Model family 2 (Amplification Hypothesis):

Model 2.1 (self-interest considered and compensatory baseline exists):

$$U\left(D\right) = \theta * \left(10 - D\right) - \left(1 - \theta\right) * |\kappa * H * R + \eta - D|$$

where *R* is the level of responsibility (*R*=5 – W). Model 2.1 is the full model in model family 2, whereas the following Model 2.2, Model 2.3, and Model 2.4 are simplified versions.

Model 2.2 (self-interest considered and compensatory baseline doesn't exist):

$$U\left(D\right) = \theta * \left(10 - D\right) - \left(1 - \theta\right) * |\kappa * H * R - D|$$

Model 2.3 (self-interest not considered and compensatory baseline exists):

$$U\left(D\right) = -|\kappa * H * R + \eta - D|$$

Model 2.4 (self-interest not considered and compensatory baseline doesn't exist):

$$U\left(D\right) = -|\kappa * H * R - D|$$

We note that Model Families 1 and 2 reflect distinct psychological mechanisms and differ fundamentally in their mathematical formulations ($\frac{H}{W}$ vs. $H * R$). Because the correlation between these two mathematical expressions is substantially below 1, model comparisons can thus empirically determine which model more accurately captures participants' compensatory decision-making.

For all the models, we transformed the trial-by-trial utility into choice probabilities (*P*) using the softmax function:

$$P\left(D\right) = \frac{e^{\lambda * U(D)}}{\sum_{j \in J} e^{\lambda * U(D_j)}}$$

where $D_j$ represents a possible number of tokens that distributed to the receiver by participants (from 0 to 10 tokens, in increments of 1). *J* is the full set of *j*. $\lambda$ is the inverse temperature parameter indicating the level of stochasticity in the compensatory decision.

## Parameter estimation

For each model, we used the fmincon function in MATLAB to estimate the parameters for each participant. To decrease the possibility of the model converging on a local minimum, we constructed a

coarse grid in the parameter space to choose the best start location. Maximum likelihood estimation was performed by maximizing the log likelihood function over each participant $i$ and trial $t$:

$$\sum_i \sum_t \log \left( P_{i,t} \left( D \right) \right)$$

## Model comparison

We compared the models adopting the Bayesian information criteria, which considers both goodness of fit and model complexity. Model 1.3 was selected as the winning model. It assumes that individuals neglect their self-interest, have a compensatory baseline, and integrate harm and the number of wrongdoers in the form of a quotient.

## Model validation

We performed a series of analyses to validate the winning model (Model 1.3). To evaluate predictive accuracy, we used the estimated parameters of the winning model to simulate each participant's compensatory decisions 50 times, generating 50 sets of simulated data. Predictive accuracy for each participant was calculated as the proportion of simulated decisions that matched their real decisions. We tested whether the predictive accuracies of our participants exceeded the chance level (9%) using a one-sample $T$ test. We also visualized the effects of harm level and responsibility level on both simulated and real decisions, checking whether the behavioral patterns of the simulated decision aligned with those of the real decisions (*Figure 3A and B*).

To demonstrate that the parameters of the winning model are identifiable, we conducted parameter recovery analyses. We refitted the model using the 50 sets of simulated data. We then evaluated the degree to which the parameter could be recovered by calculating the Pearson correlation coefficients between the parameters estimated from the real data and the parameters estimated from the simulated data.

To explore the relationships among the parameters of the winning model, we calculated the Pearson correlation coefficients among the parameters estimated from the real data.

## fMRI analyses

### fMRI data preprocessing

Imaging data were preprocessed using the Statistical Parametric Mapping 12 (SPM 12, https://www.fil.ion.ucl.ac.uk/spm/software/spm12/). Functional images were realigned to the first volume for head-motion correction, corrected for geometric distortions using field maps, slice-time corrected to the acquisition time of the middle slice. The mean functional image was co-registered with each participant's T1-weighted image using parameters from the segmentation performed on both types of images. The functional images were then normalized into the Montreal Neurological Institute (MNI) space, resampled to 3 mm isotropic voxels, and smoothed with an isotropic 8 mm full width at half-maximum Gaussian kernel.

### Activation analysis (neural representation of cognitive antecedents)

We built several separate GLMs to explore how the cognitive antecedents of guilt and shame are represented in the brain during outcome evaluation. They were designed to test different possibilities while addressing the problems of multicollinearity among regressors.

GLM 1 aimed to identify brain areas responding parametrically to the level of harm and level of responsibility. At the first level, we modeled the periods of dots (duration: 1.5 s), dots estimation (duration: 2.5 s), cue (duration: 1–3 s), outcome evaluation (duration: 4.5 s), decision making (duration: response time), decision feedback (duration: 6 s - response time), and missing trials (if existed; *Figure 1B*). The period of outcome evaluation was modeled using two event regressors: one for experimental trials where participants made incorrect dot estimations (i.e. the self-error condition), and another for filler trials where participants made correct dot estimations. All other periods were modeled using a single regressor that encompassed all trials. Our analysis focused on the period of outcome evaluation in the self-error condition. For this corresponding event regressor, we included two parametric modulators: the level of harm (H) and the number of wrongdoers (W) (number of wrongdoers = 5 – responsibility level). Notably, in our experimental design, we ensured that the

values of these two modulators were orthogonal, effectively avoiding multicollinearity issues. No additional orthogonalization adjustments were applied during the fMRI analysis. The regressors were convolved with a canonical hemodynamic response function. To control a motion effect of no interest, six head movement parameters from the realignment procedure were involved in the GLM as nuisance regressors. The inter-stimulus interval (i.e. jitter), which was not modeled in the GLM, served as an implicit baseline. We defined two contrasts at the first level, corresponding to each parametric modulator: (1) the level of harm, (2) the number of wrongdoers, each compared to the implicit baseline.

As complement analyses, we built another two GLMs to test the extent to which including the level of harm and the number of wrongdoers in the separate GLMs affected the neural findings compared to modeling them in the same GLM. The new GLMs (GLM 2 and GLM 3) were identical to GLM 1, except that each of them had only one parametric modulator (GLM 2: the level of harm; GLM 3: the number of wrongdoers) instead of two (GLM 1: the level of harm and the number of wrongdoers). For each of GLMs 2 and 3, we defined one contrast at the first level, with the parametric modulator compared to the implicit baseline.

GLM 4 targeted to identify brain areas responding to the average harm for which the participant is responsible (i.e. the quotient of the level of harm divided by the number of wrongdoers). It is the same as the GLM 3, except that the parametric modulator was replaced with the quotient ($\frac{H}{W}$). For GLM 4, we defined one contrast at the first level, with the parametric modulator compared to the implicit baseline.

GLM 5 set out to identify brain areas responding to the product of the level of harm and the level of responsibility (R). It is the same as the GLM 4, except that the parametric modulator was replaced with the product (H×R). For GLM 5, we defined one contrast at the first level, with the parametric modulator compared to the implicit baseline.

At the second level, we submitted the first-level contrast images from all the GLMs above to one-sample $T$ tests. The statistical threshold was set as $p<0.05$ [family-wise error correction (FWE) at the cluster level] with an initial cluster-defining voxel-level threshold of $p<0.001$ for the whole-brain analysis. This statistical threshold was applied to all fMRI data analyses unless stated otherwise.

## Activation analysis (neural basis of emotion sensitivity)

The harm-driven guilt sensitivity, responsibility-driven guilt sensitivity, harm-driven shame sensitivity, and responsibility-driven shame sensitivity refer to the degree to which harm and responsibility elicit guilt and shame for an individual. We quantified them by the individual-specific coefficient estimates of harm and responsibility on guilt and shame from the linear mixed-effect regressions (*Appendix 3—table 3* and *Appendix 3—table 4*). To examine whether these emotion sensitivities are correlated with neural responses to the quotient of harm divided by the number of wrongdoers, we submitted the first-level contrast images from GLM 4 to one-sample $T$ tests and involved the four types of emotion sensitivity as covariates at the second level.

The results showed that the neural responses in the temporoparietal junction/superior temporal sulcus (TPJ/STS) and precentral cortex/postcentral cortex/supplementary motor area (PRC/POC/SMA) were negatively correlated with the responsibility-driven shame sensitivity. To test whether these regions were more involved in responsibility-driven shame sensitivity than in other types of emotion sensitivity, we implemented a leave-one-subject-out (LOSO) cross-validation procedure (e.g. *Esterman et al., 2010*). In each fold, clusters in the TPJ/STS and PRC/POC/SMA showing significant correlations with responsibility-driven shame sensitivity were identified at the group level based on $N$-1 participants. These clusters, defined as regions of interest (ROI), were then applied to the left-out participant, from whom we extracted the mean parameter estimates (i.e. neural response values). If, in a given fold, no suprathreshold cluster was detected within the TPJ/STS or PRC/POC/SMA after correction, or if the two regions merged into a single cluster that could not be separated, the corresponding value was coded as missing. Repeating this procedure across all folds yielded an independent set of ROI-based estimates for each participant. In the LOSO cross-validation procedure, the TPJ/STS and PRC/POC/SMA merged into a single inseparable cluster in two folds, and no suprathreshold cluster was detected within the TPJ/STS in one fold. These instances were coded as missing, resulting in valid data from 39 participants for the TPJ/STS and 40 participants for the PRC/POC/SMA. We then correlated these estimates with all four types of emotion sensitivities and compared the

correlation with responsibility-driven shame sensitivity against those with the other sensitivities using *Z* tests (Pearson and Filon's *Z*).

## Activation analysis (neural basis of compensatory sensitivity)

The guilt-driven compensatory sensitivity and shame-driven compensatory sensitivity refer to the extent to which guilt and shame are converted into compensation for an individual. They can be indicated by the individual-specific coefficient estimates of guilt and shame on compensation from the linear mixed-effect regression (*Appendix 3—table 7*). To identify whether these two types of compensatory sensitivity are associated with brain activity during outcome evaluation, an additional contrast was defined at the first level based on GLM 4. This contrast examined average brain activity during outcome evaluation (effect of the event regressor: viewing the outcome in the self-error condition) (*Losin et al., 2020*). At the second level, we submitted the first-level contrast images from GLM 4 to one-sample *T* tests and involved the two types of compensatory sensitivity as covariates.

We found that neural activity in the left TP, bilateral IPL, and left lateral prefrontal cortex (LPFC) was only correlated with guilt-driven or shame-driven compensatory sensitivity (*Figure 5A and B*). To directly test whether these regions were more involved in one of the two types of compensatory sensitivity, we applied the same LOSO cross-validation procedure described above. In this procedure, no suprathreshold cluster was detected within the LPFC in one fold and within the TP in 27 folds. These cases were coded as missing, resulting in valid data from 42 participants for the bilateral IPL, 41 participants for the LPFC, and 15 participants for the TP. The limited sample size for the TP likely reflects that its effect was only marginally above the correction threshold, such that the reduced power in cross-validation often rendered it nonsignificant. Because the sample size for the TP was too small and the results may therefore be unreliable, we did not pursue further analyses for this region. The independent ROI-based estimates were then correlated with both guilt-driven and shame-driven compensatory sensitivities, and the strength of the correlations was compared using *Z* tests (Pearson and Filon's *Z*).

Compensation may be driven by emotion or motivation other than guilt and shame (e.g. indebtedness; *Gao et al., 2024*). The parameter $\kappa$ from our winning computational model (i.e. Model 1.3) is supposed to capture the combined influence of various psychological processes on compensation, including guilt and shame. To confirm the relationships between $\kappa$ and guilt-driven and shame-driven compensatory sensitivities, we conducted a linear mixed-effects regression. $\kappa$ was regressed onto guilt-driven and shame-driven compensatory sensitivities, with participant-specific random intercepts and random slopes for each fixed effect included as random effects. Besides, to more comprehensively explore the neural basis of compensatory sensitivity, we submitted the first-level contrast images from GLM 4 to one-sample *T* tests, incorporating the values of $\kappa$ as covariates.

## Activation analysis and mediation analysis (neural correlates of trait guilt and compensation)

As we observed significant correlations between trait guilt scores and compensatory behavior, we further explored brain activity that mediates their relationships. At the first step, we attempted to identify the neural correlates of the two dimensions of trait guilt by submitting first-level contrast images from GLM 4 to one-sample *T* tests and involved the negative behavior-evaluations and repair action tendencies scores as covariates at the second level. The aMCC and midbrain nuclei are defined as the ROIs, given that they have been found to be associated with guilt and compensation (*Yu et al., 2014*). The ROIs for aMCC and midbrain nuclei were respectively defined as a 6-mm-radius sphere centered at the Montreal Neurological Institute (MNI) coordinates [0, 34, 16] and [-2,–20, –20] (*Yu et al., 2014*). Brain activities that revealed significant correlations with trait guilt scores were considered as candidate mediators (e.g., *Losin et al., 2020*). The statistical threshold was set as $p < 0.05$ [family-wise error correction (FWE) at the cluster level] with an initial cluster-defining voxel-level threshold of $p < 0.001$ for the small-volume correction analysis or the whole-brain analysis.

At the second step, we examined whether the candidate mediators mediated the relationship between trait guilt and compensation using the PROCESS macro based on the SPSS software (http://www.processmacro.org/index.html). We tested the mediation effect, with participants' trait guilt scores as the predictor variable, compensation as the outcome variable, and mean estimates extracted from the clusters we identified in the first stage or the ROIs (*Appendix 3—table 18*) as the

mediator variable (one candidate mediator per mediation analysis). We used a bootstrap procedure (randomly sampling 5000 observations with replacement) to obtain 95% confidence intervals (CI) of path coefficients for significance testing. If the CI of the path coefficient did not cover zero, the effect was considered as significant.

For completeness, we explored the neural correlates of other traits by submitting first-level contrast images from GLM 4 to one-sample $T$ tests and involved the trait shame, trait gratitude, or SVO scores as covariates at the second level.

## Acknowledgements

This project has received funding from the Scientific and Technological Innovation (STI) 2030-Major Projects (2021ZD0200500 to Chao Liu), the National Natural Science Foundation of China (32200884 to Ruida Zhu; 32441109, 32271092, 32130045 to Chao Liu), the Open Research Fund of the State Key Laboratory of Cognitive Neuroscience and Learning (CNLYB2404 to Ruida Zhu and Chao Liu), the Start-up Project for Support of Young Doctors (SL2023A04J00351 to Ruida Zhu), the Fundamental Research Funds for the Central Universities (Sun Yat-sen University; 2024qntd90 to Ruida Zhu), the Beijing Major Science and Technology Project under Contract No. Z241100001324005 (to Chao Liu), and the Opening Project of the State Key Laboratory of General Artificial Intelligence (SKLAGI20240P06 to Chao Liu).

## Additional information

### Competing interests

Yi Zeng: Yi Zeng is affiliated with Long-term AI, China. The author has no other competing interests to declare. The other authors declare that no competing interests exist.

### Funding

| Funder | Grant reference number | Author |
|---|---|---|
| Scientific and Technological Innovation (STI) 2030-Major Projects | 2021ZD0200500 | Chao Liu |
| National Natural Science Foundation of China | 32200884 | Ruida Zhu |
| National Natural Science Foundation of China | 32441109 | Chao Liu |
| National Natural Science Foundation of China | 32271092 | Chao Liu |
| National Natural Science Foundation of China | 32130045 | Chao Liu |
| Open Research Fund of the State Key Laboratory of Cognitive Neuroscience and Learning | CNLYB2404 | Ruida Zhu Chao Liu |
| Start-up Project for Support of Young Doctors | SL2023A04J00351 | Ruida Zhu |
| Fundamental Research Funds for the Central Universities | Sun Yat-sen University, 2024qntd90 | Ruida Zhu |
| Beijing Major Science and Technology Project | Z241100001324005 | Chao Liu |
| Opening Project of the State Key Laboratory of General Artificial Intelligence | SKLAGI20240P06 | Chao Liu |

| Funder | Grant reference number | Author |
| --- | --- | --- |

The funders had no role in study design, data collection and interpretation, or the decision to submit the work for publication.

## Author contributions
Ruida Zhu, Conceptualization, Formal analysis, Funding acquisition, Methodology, Writing – original draft, Writing – review and editing; Huanqing Wang, Investigation, Writing – review and editing; Chunliang Feng, Linyuan Yin, Ran Zhang, Yi Zeng, Writing – review and editing; Chao Liu, Conceptualization, Supervision, Funding acquisition, Writing – original draft, Writing – review and editing

## Author ORCIDs
Ruida Zhu ⓘD https://orcid.org/0000-0002-1316-7526
Chao Liu ⓘD https://orcid.org/0000-0003-1149-2314

## Ethics
Participants provided written consent. The experimental protocol (protocol number: ICBIR_A_0071_010) was approved by the local research ethics committee at the State Key Laboratory of Cognitive Neuroscience and Learning, Beijing Normal University.

Reviewer #1 (Public review): https://doi.org/10.7554/eLife.107223.3.sa1
Reviewer #2 (Public review): https://doi.org/10.7554/eLife.107223.3.sa2
Reviewer #3 (Public review): https://doi.org/10.7554/eLife.107223.3.sa3
Author response https://doi.org/10.7554/eLife.107223.3.sa4

# Additional files

## Supplementary files
MDAR checklist

## Data availability
The code, behavioral data, and preprocessed fMRI data are public available on the Open Science Framework (https://osf.io/sve7h/files).

The following dataset was generated:

| Author(s) | Year | Dataset title | Dataset URL | Database and Identifier |
| --- | --- | --- | --- | --- |
| Zhu R | 2025 | Human neurocomputational mechanisms of guilt-driven and shame-driven altruistic behavior | https://osf.io/sve7h | Open Science Framework, sve7h |

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

## Appendix 1

### Illustrative examples of the theoretical relationships among harm, responsibility, guilt, and shame within the framework of functionalist theories

#### Based on functionalist theories, guilt functions to minimize and repair undue harm on valued others, thereby addressing the adaptive problem of insufficiently valuing others

Imagine that you attempt to rescue your roommate from a fire but fail to extinguish it within the optimal time. The fire eventually burns out on its own. You would likely feel greater guilt if your roommate suffered severe burns resulting in permanent damage (i.e. severe harm) than if they only experienced minor burns from which they could fully recover (i.e. minor harm). This is because guilt should be sensitive to the severity of harm inflicted on valued others, if it serves to decrease the inequality caused by the harm. Since the severity of harm is situationally determined and random (at least to some extent), you might believe that others would not fully base their devaluation of you on this factor. Consequently, the influence of harm on shame may be smaller than its influence on guilt.

#### Shame functions to prevent and mitigate the threat of being devalued by others, thus addressing the adaptive problem of reputational damage to the self

In another example, you attempt to rescue your roommate from a fire alongside others. If any one of you fails to fetch enough water in time, the fire cannot be extinguished within the optimal time. The firefighting task failed resulting in your roommate getting burned. You would likely feel greater shame if you were the only one who failed to fetch enough water to extinguish the fire (i.e. full responsibility) than if all of you failed to fetch enough water (i.e. partial responsibility). This is because the responsibility you take can expose deficiencies in your morality (willingness to help) and/or competence (physical ability), leading to devaluation by others, which shame monitors and reflects. Since the degree of responsibility is not directly related to the severity of harm that guilt tracks, the influence of responsibility on guilt may be smaller than its influence on shame.

## Appendix 2

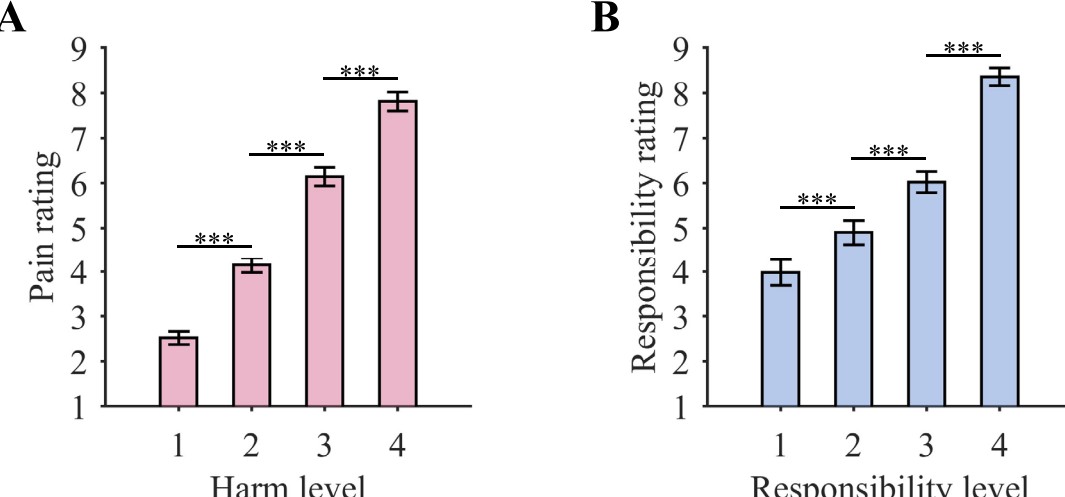

**Appendix 2—figure 1.** Manipulation checks. (**A**) The differences in pain ratings across different harm levels were all significant. (**B**) The differences in responsibility ratings across different responsibility levels were all significant. ***p < 0.001.

# Appendix 3

**Appendix 3—table 1.** Means (and standard deviations) of participants' pain ratings across different harm levels.

| Conditions | Pain ratings |
|---|---|
| Harm level 1 | 2.52 (0.94) |
| Harm level 2 | 4.14 (1.07) |
| Harm level 3 | 6.14 (1.34) |
| Harm level 4 | 7.81 (1.38) |

**Appendix 3—table 2.** Means (and standard deviations) of participants' responsibility ratings across different responsibility levels.

| Conditions | Responsibility ratings |
|---|---|
| Making correct estimates | 2.02 (1.37) |
| Responsibility level 1 | 3.98 (1.89) |
| Responsibility level 2 | 4.90 (1.74) |
| Responsibility level 3 | 6.02 (1.54) |
| Responsibility level 4 | 8.36 (1.28) |

**Appendix 3—table 3.** Linear mixed-effect regression results for emotion ratings by harm and emotion type.

| Fixed-effect regressor | Estimate | SE | T (df) | p |
|---|---|---|---|---|
| Harm | 0.231 | 0.083 | 2.77 (67) | 0.007 |
| Emotion type | −1.259 | 0.327 | 3.85 (41) | <0.001 |
| Harm ×Emotion type | 0.512 | 0.086 | 5.96 (41) | <0.001 |
| Intercept | 3.738 | 0.358 | 10.45 (61) | <0.001 |

Note: Dependent variable: emotion ratings; *SE*: standard error; *df*: degrees of freedom; Regarding emotion type, 'shame' was defined as the reference category.

**Appendix 3—table 4.** Linear mixed-effect regression results for emotion ratings by responsibility and emotion type.

| Fixed-effect regressor | Estimate | SE | T (df) | p |
|---|---|---|---|---|
| Responsibility | 0.927 | 0.109 | 8.54 (61) | <0.001 |
| Emotion type | 0.903 | 0.366 | 2.47 (41) | 0.018 |
| Responsibility ×Emotion type | −0.355 | 0.099 | 3.60 (41) | <0.001 |
| Intercept | 1.996 | 0.365 | 5.47 (66) | <0.001 |

Note: Dependent variable: emotion ratings; *SE*: standard error; *df*: degrees of freedom; Regarding emotion type, 'shame' was defined as the reference category.

**Appendix 3—table 5.** Linear mixed-effect regression results for emotion ratings by harm, responsibility, and emotion type.

| Fixed-effect regressor | Estimate | SE | T (df) | p |
|---|---|---|---|---|
| Harm | −0.079 | 0.136 | 0.58 (78) | 0.564 |
| Responsibility | 0.615 | 0.167 | 3.69 (74) | <0.001 |
| Emotion type | −0.157 | 0.576 | 0.274 (68) | 0.785 |

*Appendix 3—table 5 Continued on next page*

*Appendix 3—table 5 Continued*

| Fixed-effect regressor | Estimate | SE | T (df) | p |
|---|---|---|---|---|
| Harm ×Responsibility | 0.124 | 0.052 | 2.39 (81) | 0.019 |
| Harm ×Emotion type | 0.423 | 0.171 | 2.47 (68) | 0.016 |
| Responsibility ×Emotion type | –0.443 | 0.194 | 2.29 (63) | 0.026 |
| Harm ×Responsibility × Emotion type | 0.036 | 0.069 | 0.53 (71) | 0.601 |
| Intercept | 2.198 | 0.434 | 5.06 (81) | <0.001 |

Note: Dependent variable: emotion ratings; *SE*: standard error; *df*: degrees of freedom; Regarding emotion type, 'shame' was defined as the reference category.

**Appendix 3—table 6.** Linear mixed-effect regression models of compensation.

| Model | Dependent variable | Fixed-effect regressors | BIC value |
|---|---|---|---|
| I | Compensation | Guilt | 9136 |
| II | Compensation | Shame | 9420 |
| III | Compensation | Guilt and shame | 9116* |
| IV | Compensation | Guilt, shame, and guilt ×shame | 9147 |

Note: BIC, Bayesian information criterion.
*Best model.

**Appendix 3—table 7.** Linear mixed-effect regression results for compensation by guilt and shame.

| Fixed-effect regressor | Estimate | SE | T (df) | p |
|---|---|---|---|---|
| Guilt | 0.215 | 0.042 | 5.16 (41) | <0.001 |
| Shame | 0.095 | 0.026 | 3.69 (36) | <0.001 |
| Intercept | 2.988 | 0.316 | 9.47 (43) | <0.001 |

Note: Dependent variable: compensation; *SE*: standard error; *df*: degrees of freedom.

**Appendix 3—table 8.** Model comparison for compensatory behavior.

| Model | Brief model description | BIC values |
|---|---|---|
| Model 1.1 | Integration in the form of a quotient, self-interest, and compensation baseline | 8095 |
| Model 1.2 | Integration in the form of a quotient and self-interest | 10566 |
| Model 1.3 | Integration in the form of a quotient and compensation baseline | 8001* |
| Model 1.4 | Integration in the form of a quotient | 10408 |
| Model 2.1 | Integration in the form of a product, self-interest, and compensation baseline | 8138 |
| Model 2.2 | Integration in the form of a product and self-interest | 10472 |
| Model 2.3 | Integration in the form of a product and compensation baseline | 8046 |
| Model 2.4 | Integration in the form of a product | 10314 |

Note: BIC, Bayesian information criterion.
*Best model.

**Appendix 3—table 9.** Brain regions responding parametrically to the level of harm and the number of wrongdoers during outcome evaluation (results based on GLM 1).

| Modulator/Region | Cluster Size | Peak MNI Coordinates | | | T | $P_{FWE}$ |
|---|---|---|---|---|---|---|
| | | x | y | z | | |
| *Modulator: H* | | | | | | |

*Appendix 3—table 9 Continued on next page*

*Appendix 3—table 9 Continued*

| Modulator/Region | Cluster Size | Peak MNI Coordinates | | | *T* | *P*~FWE~ |
|---|---|---|---|---|---|---|
| None | | | | | | |
| *Modulator: W* | | | | | | |
| Precentral and postcentral cortex | 484 | –21 | –27 | 66 | 4.70 | <0.001 |
| Precentral and postcentral cortex | 168 | 21 | –33 | 66 | 4.23 | 0.023 |

Note: H denotes the level of harm; W denotes the number of wrongdoers; MNI, Montreal Neurological Institute; FWE, family-wise error correction.

**Appendix 3—table 10.** Brain regions responding parametrically to the level of harm and the number of wrongdoers during outcome evaluation (results based on GLMs 2 and 3).

| Modulator/Region | Cluster Size | Peak MNI Coordinates | | | *T* | *P*~FWE~ |
|---|---|---|---|---|---|---|
| | | x | y | z | | |
| *Modulator: H* | | | | | | |
| None | | | | | | |
| *Modulator: W* | | | | | | |
| Precentral and postcentral cortex | 514 | –21 | –27 | 66 | 4.70 | <0.001 |
| Precentral and postcentral cortex | 171 | 21 | –33 | 66 | 4.22 | 0.022 |

Note: H denotes the level of harm; W denotes the number of wrongdoers; MNI, Montreal Neurological Institute; FWE, family-wise error correction.

**Appendix 3—table 11.** Brain regions responding parametrically to the quotient of the level of harm divided by the number of wrongdoers during outcome evaluation (results based on GLM 4).

| Modulator/Region | Cluster Size | Peak MNI Coordinates | | | *T* | *P*~FWE~ |
|---|---|---|---|---|---|---|
| | | x | y | z | | |
| *Modulator: $\frac{H}{W}$* | | | | | | |
| Striatum | 545 | 24 | -6 | 21 | –6.84 | <0.001 |
| Posterior insula | 181 | –48 | -3 | 15 | –4.43 | 0.021 |

Note: $\frac{H}{W}$ denotes the quotient of the level of harm divided by the number of wrongdoers. MNI, Montreal Neurological Institute; FWE, family-wise error correction. Negative *T* values indicate that the activity in the brain region is inversely related to the modulator.

**Appendix 3—table 12.** Neural activity in the brain regions correlated with the parameter estimates of responsibility-driven shame sensitivity.

| *Covariate*/Region | Cluster Size | Peak MNI Coordinates | | | *T* | *P*~FWE~ |
|---|---|---|---|---|---|---|
| | | x | y | z | | |
| *Covariate: responsibility-driven shame sensitivity* | | | | | | |
| TPJ/STS | 386 | –69 | –24 | 12 | 4.68 | 0.001 |
| PRC/POC/SMA | 2372 | 15 | –30 | 75 | 6.27 | <0.001 |

Note: TPJ, temporoparietal junction; STS, superior temporal sulcus; PRC, precentral cortex; POC, postcentral cortex; SMA, supplementary motor area. MNI, Montreal Neurological Institute; FWE, family-wise error correction.

**Appendix 3—table 13.** The differences in correlations between the brain activities and different types of emotion sensitivity.

| Brain region | Emotion sensitivity | Correlation 1 | | Correlation 2 | | Correlation Difference | |
|---|---|---|---|---|---|---|---|
| | | R | p | R | p | Z | p |
| TPJ/STS | RDSS vs. RDGS | –0.51 | 0.001 | 0.04 | 0.797 | 2.44 | 0.015 |
| TPJ/STS | RDSS vs. HDSS | –0.51 | 0.001 | 0.21 | 0.211 | 3.38 | <0.001 |
| TPJ/STS | RDSS vs. HDGS | –0.51 | 0.001 | –0.17 | 0.308 | 1.87 | 0.062 |
| PRC/POC/SMA | RDSS vs. RDGS | –0.68 | <0.001 | 0.20 | 0.206 | 4.36 | <0.001 |
| PRC/POC/SMA | RDSS vs. HDSS | –0.68 | <0.001 | 0.12 | 0.457 | 4.17 | <0.001 |
| PRC/POC/SMA | RDSS vs. HDGS | –0.68 | <0.001 | –0.15 | 0.354 | 3.30 | 0.001 |

Note: TPJ, temporoparietal junction; STS, superior temporal sulcus; PRC, precentral cortex; POC, postcentral cortex; SMA, supplementary motor area; RDSS, responsibility-driven shame sensitivity; RDGS, responsibility-driven guilt sensitivity; HDSS, harm-driven shame sensitivity; HDGS, harm-driven guilt sensitivity. During the leave-one-subject-out cross-validation procedure, TPJ/STS and PRC/POC/SMA merged into a single inseparable cluster in two folds, and no suprathreshold cluster was detected within TPJ/STS in another fold. These cases were coded as missing, resulting in 39 participants for TPJ/STS and 40 participants for PRC/POC/SMA.

**Appendix 3—table 14.** Neural activity in the brain regions correlated with the parameter estimates of guilt-driven or shame-driven compensatory sensitivities.

| Covariate/Region | Cluster Size | Peak MNI Coordinates | | | T | P_FWE |
|---|---|---|---|---|---|---|
| | | x | y | z | | |
| *Covariate: guilt-driven compensatory sensitivity* | | | | | | |
| DMPFC/SMA | 394 | 12 | 21 | 57 | 5.02 | <0.001 |
| TP | 220 | –57 | 6 | –27 | 4.99 | 0.005 |
| lIPL | 131 | –36 | –60 | 54 | 4.44 | 0.036 |
| *Covariate: shame-driven compensatory sensitivity* | | | | | | |
| DMPFC/SMA | 783 | -9 | 21 | 48 | 5.28 | <0.001 |
| lLPFC | 234 | –33 | 51 | 12 | 4.54 | 0.003 |
| rIPL | 285 | 36 | –54 | 54 | 5.32 | 0.001 |
| lIPL | 283 | –33 | –51 | 39 | 5.25 | 0.001 |

Note: DMPFC, dorsomedial prefrontal cortex; LPFC, lateral prefrontal cortex; SMA, supplementary motor area; TP, temporal pole; IPL, inferior parietal lobe; ITC, inferior temporal cortex. r, right; l, left; MNI, Montreal Neurological Institute; FWE, family-wise error correction.

**Appendix 3—table 15.** The differences in correlations between the brain activity and different types of compensatory sensitivity.

| Brain region | Emotion sensitivity | Correlation 1 | | Correlation 2 | | Correlation Difference | |
|---|---|---|---|---|---|---|---|
| | | R | p | R | p | Z | p |
| rIPL | GDCS vs. SDCS | 0.43 | 0.005 | 0.56 | <0.001 | 1.01 | 0.312 |
| lIPL | GDCS vs. SDCS | 0.42 | 0.005 | 0.47 | 0.002 | 0.37 | 0.709 |
| lLPFC | GDCS vs. SDCS | 0.15 | 0.335 | 0.44 | 0.004 | 1.93 | 0.053 |

Note: IPL, inferior parietal lobe; LPFC, lateral prefrontal cortex; r, right; l, left; GDCS, guilt-driven compensatory sensitivity; SDCS, shame-driven compensatory sensitivity.

**Appendix 3—table 16.** Linear regression results for parameter $\kappa$ by guilt-driven and shame-driven compensatory sensitivities.

| Fixed-effect regressor | Estimate | SE | T (df) | p |
|---|---|---|---|---|
| Guilt-driven compensatory sensitivity | 2.062 | 0.352 | 5.85 (38) | <0.001 |
| Shame-driven compensatory sensitivity | 2.426 | 0.865 | 2.82 (38) | 0.008 |
| Intercept | –0.048 | 0.107 | –0.45 (38) | 0.653 |

Note: Dependent variable: compensation; SE: standard error; df: degrees of freedom.

**Appendix 3—table 17.** Neural activity in the brain regions correlated with the parameter estimates of $\kappa$.

| Covariate/Region | Cluster Size | Peak MNI Coordinates | | | T | $P_{FWE}$ |
|---|---|---|---|---|---|---|
| | | x | y | z | | |
| *Covariate: $\kappa$* | | | | | | |
| DMPFC/SMA/LPFC | 1702 | 54 | 36 | 27 | 5.70 | <0.001 |
| aINS | 173 | 33 | 18 | 6 | 4.98 | 0.013 |
| TP/aINS | 771 | –54 | 9 | –30 | 5.31 | <0.001 |
| rIPL | 283 | 36 | –54 | 51 | 5.39 | 0.001 |
| lIPL | 249 | –33 | –57 | 57 | 5.22 | 0.003 |
| ITC | 215 | 57 | –48 | –18 | 5.86 | 0.005 |

Note: DMPFC, dorsomedial prefrontal cortex; SMA, supplementary motor area; LPFC, lateral prefrontal cortex; aINS, anterior insula; TP, temporal pole; IPL, inferior parietal lobe; ITC, inferior temporal cortex. r, right; l, left; MNI, Montreal Neurological Institute; FWE, family-wise error correction.

**Appendix 3—table 18.** The brain regions whose responses to the quotient of harm divided by the number of wrongdoers correlated with repair action tendencies (i.e., scores of a trait guilt subscale).

| Covariate/Region | Cluster Size | Peak MNI Coordinates | | | T | $P_{FWE}$ |
|---|---|---|---|---|---|---|
| | | x | y | z | | |
| *Covariate: repair action tendencies (trait guilt)* | | | | | | |
| aMCC[*] | 12 | -3 | 30 | 15 | 4.21 | 0.003 |
| aMCC/DMPFC/SMA/precuneus | 4003 | 3 | 54 | 24 | 5.55 | <0.001 |
| LPFC | 166 | 33 | 27 | 36 | 4.37 | 0.021 |
| aINS | 418 | –30 | 15 | -9 | 4.93 | <0.001 |
| rTP/ITC | 182 | 48 | –12 | –27 | 4.69 | 0.015 |
| lTP/ITC | 241 | –42 | -6 | –33 | 4.67 | 0.005 |
| IPL/TPJ | 141 | 51 | –39 | 18 | 4.63 | 0.035 |

Note: aMCC, anterior middle cingulate cortex; DMPFC, dorsomedial prefrontal cortex; SMA, supplementary motor area; LPFC, lateral prefrontal cortex; aINS, anterior insula; TP, temporal pole; ITC, inferior temporal cortex; IPL, inferior parietal lobe; TPJ, temporoparietal junction. r, right; l, left; MNI, Montreal Neurological Institute; FWE, family-wise error correction. [*]small-volume correction.

**Appendix 3—table 19.** The number of filler trials with different outcomes.

| | Pain levels | | | | | |
|---|---|---|---|---|---|---|
| Number of filler trials (48 in total) | No pain stimulation | 1 | 2 | 3 | 4 |
| Number of wrong deciders excluding the participant | | | | | |
| 0 | 24 | - | - | - | - |

*Appendix 3—table 19 Continued on next page*

*Appendix 3—table 19 Continued*

| Pain levels | | | | | |
|---|---|---|---|---|---|
| 1 | - | 3 | 3 | 3 | 3 |
| 2 | - | 2 | 2 | 2 | 2 |
| 3 | - | 1 | 1 | 1 | 1 |

