## [Editor Report · eLife Assessment]

This is an **important** study on how dissociable emotions of shame and guilt emerge from cognitive processes and guide behavioral responses. The task is well designed and yields **compelling** behavioral, computational, and neural evidence elucidating the cognitive link between emotions and compensatory decisions. The work has broad theoretical and practical implications across a range of disciplines concerned with human behavior, including psychology, neuroscience, economics, public policy, and psychiatry.

---

## [Referee Report · Reviewer #1 (Public review)]

This work provides important new evidence of the cognitive and neural mechanisms that give rise to feelings of shame and guilt, as well as their transformation into compensatory behavior. The authors use a well-designed interpersonal task to manipulate responsibility and harm, eliciting varying levels of shame and guilt in participants. The study combines behavioral, computational, and neuroimaging approaches to offer a comprehensive account of how these emotions are experienced and acted upon. Notably, the findings reveal distinct patterns in how harm and responsibility contribute to guilt and shame and how these factors are integrated into compensatory decision-making.

Strengths:

• Investigating both guilt and shame in a single experimental framework allows for a direct comparison of their behavioral and neural effects while minimizing confounds

• The study provides a novel contribution to the literature by exploring the neural bases underlying the conversion of shame into behavior

• The task is creative and ecologically valid, simulating a realistic social situation while retaining experimental control

• Computational modeling and fMRI analysis yield converging evidence for a quotient-based integration of harm and responsibility in guiding compensatory behavior

Limitations:

The authors address the study's limitations and offer well-reasoned explanations for their methodological choices.

The conclusions of the paper are well supported by the data. It would be valuable for future studies to validate these findings using alternative tasks or paradigms, to ensure the robustness and generalizability of the observed behavioral and neural mechanisms. Overall, this is a well-executed and insightful study that makes a meaningful contribution to understanding the cognitive and neural mechanisms underlying guilt and shame.

---

## [Referee Report · Reviewer #2 (Public review)]

Summary:

The authors combined behavioral experiments, computational modeling, and functional magnetic resonance imaging (fMRI) to investigate the psychological and neural mechanisms underlying guilt, shame, and the altruistic behaviors driven by these emotions. The results revealed that guilt is more strongly associated with harm, whereas shame is more closely linked to responsibility. Compared to shame, guilt elicited a higher level of altruistic behavior. Computational modeling demonstrated how individuals integrate information about harm and responsibility. The fMRI findings identified a set of brain regions involved in representing harm and responsibility, transforming responsibility into feelings of shame, converting guilt and shame into altruistic actions, and mediating the effect of trait guilt on compensatory behavior.

Strengths:

This study offers a significant contribution to the literature on social emotions by moving beyond prior research that typically focused on isolated aspects of guilt and shame. The study presents a comprehensive examination of these emotions, encompassing their cognitive antecedents, affective experiences, behavioral consequences, trait-level characteristics, and neural correlates. The authors have introduce a novel experimental task that enables such a systematic investigation and holds strong potential for future research applications. The computational modeling procedures were implemented in accordance with current field standards. The findings are rich and offer meaningful theoretical insights. The manuscript is well written, and the results are clearly and logically presented.

Weaknesses:

In this study, participants' feelings of guilt and shame were assessed retrospectively, after they had completed all altruistic decision-making tasks. This reliance on memory-based self-reports may introduce recall bias, potentially compromising the accuracy of the emotion measurements.

In many behavioral economic models, self-interest plays a central role in shaping individual decision-making, including moral decisions. However, the model comparison results in this study suggest that models without a self-interest component (such as Model 1.3) outperform those that incorporate it (such as Model 1.1 and Model 1.2). The authors have not provided a satisfactory explanation for this counterintuitive finding.

The phrases "individuals integrate harm and responsibility in the form of a quotient" and "harm and responsibility are integrated in the form of a quotient" appear in the Abstract and Discussion sections. However, based on the results of the computational modeling, it is more accurate to state that "harm and the number of wrongdoers are integrated in the form of a quotient." The current phrasing misleadingly suggests that participants represent information as harm divided by responsibility, which does not align with the modeling results. This potentially confusing expression should be revised for clarity and accuracy.

In the Discussion, the authors state: "Since no brain region associated social cognition showed significant responses to harm or responsibility, it appears that human brain encodes a unified measure integrating harm and responsibility (i.e., the quotient) rather than processing them as separate entities when both are relevant to subsequent emotional experience and decision-making." However, this interpretation overstates the implications of the null fMRI findings. The absence of significant activation in response to harm or responsibility does not necessarily imply that the brain does not represent these dimensions separately. Null results can arise from various factors, including limitations in the sensitivity of fMRI. It is possible that more fine-grained techniques, such as intracranial electrophysiological recordings, could reveal distinct neural representations of harm and responsibility. The interpretation of these null findings should be made with greater caution.

For the revised manuscript, the authors have provided additional evidence and clarified expressions. all the comments were responded. I have no further comments.

---

## [Referee Report · Reviewer #3 (Public review)]

Summary:

Zhu et al. set out to elucidate how the moral emotions of guilt and shame emerge from specific cognitive antecedents - harm and responsibility - and how these emotions subsequently drive compensatory behavior. Consistent with their prediction derived from functionalist theories of emotion, their behavioral findings indicate that guilt is more influenced by harm, whereas shame is more influenced by responsibility. In line with previous research, their results also demonstrate that guilt has a stronger facilitating effect on compensatory behavior than shame. Furthermore, computational modeling and neuroimaging results suggest that individuals integrate harm and responsibility information into a composite representation of the individual's share of the harm caused. Brain areas such as the striatum, insula, temporoparietal junction, lateral prefrontal cortex, and cingulate cortex were implicated in distinct stages of the processing of guilt and/or shame. In general, this work makes an important contribution to the field of moral emotions. Its impact could be further enhanced by clarifying methodological details, offering a more nuanced interpretation of the findings, and discussing their potential practical implications in greater depth.

Strengths:

First, this work conceptualizes guilt and shame as processes unfolding across distinct stages (cognitive appraisal, emotional experience, and behavioral response) and investigates the psychological and neural characteristics associated with their transitions from one stage to the next.

Second, the well-designed experiment effectively manipulates harm and responsibility - two critical antecedents of guilt and shame.

Third, the findings deepen our understanding of the mechanisms underlying guilt and shame beyond what has been established in previous research.

Comments on revisions:

The authors have addressed the issues I raised in the previous review. I have no more comments on the manuscript.

---

## [Author Response]

The following is the authors’ response to the original reviews.

**Reviewer #1 (Public review):**
SummaryThis work provides important new evidence of the cognitive and neural mechanisms that give rise to feelings of shame and guilt, as well as their transformation into compensatory behavior. The authors use a well-designed interpersonal task to manipulate responsibility and harm, eliciting varying levels of shame and guilt in participants. The study combines behavioral, computational, and neuroimaging approaches to offer a comprehensive account of how these emotions are experienced and acted upon. Notably, the findings reveal distinct patterns in how harm and responsibility contribute to guilt and shame and how these factors are integrated into compensatory decision-making.Strengths(1) Investigating both guilt and shame in a single experimental framework allows for a direct comparison of their behavioral and neural effects while minimizing confounds.(2) The study provides a novel contribution to the literature by exploring the neural bases underlying the conversion of shame into behavior.(3) The task is creative and ecologically valid, simulating a realistic social situation while retaining experimental control.(4) Computational modeling and fMRI analysis yield converging evidence for a quotient-based integration of harm and responsibility in guiding compensatory behavior.

We are grateful for your thoughtful summary of our work’s strengths and greatly appreciate these positive words.

We would like to note that, in accordance with the journal’s requirements, we have uploaded both a clean version of the revised manuscript and a version with all modifications highlighted in blue.

Weakness(1) Post-experimental self-reports rely both on memory and on the understanding of the conceptual difference between the two emotions. Additionally, it is unclear whether the 16 scenarios were presented in random order; sequential presentation could have introduced contrast effects or demand characteristics.

Thank you for pointing out the two limitations of the experimental paradigm. We fully agree with your point. Participants recalled and reported their feelings of guilt and shame immediately after completing the task, which likely ensured reasonably accurate state reports. We acknowledge, however, that in-task assessments might provide greater precision. We opted against them to examine altruistic decision-making in a more natural context, as in-task assessments could have heightened participants’ awareness of guilt and shame and biased their altruistic decisions. Post-task assessments also reduced fMRI scanning time, minimizing discomfort from prolonged immobility and thereby preserving data quality.

In the present study, assessing guilt and shame required participants to distinguish conceptually between the two emotions. Most research with adult participants has adopted this approach, relying on direct self-reports of emotional intensity under the assumption that adults can differentiate between guilt and shame (Michl et al., 2014; Wagner et al., 2011; Zhu et al., 2019). However, we acknowledge that this approach may be less suitable for studies involving children, who may not yet have a clear understanding of the distinction between guilt and shame.

The limitations have been added into the Discussion section (Page 47): “This research has several limitations. First, post-task assessments of guilt and shame, unlike in-task assessments, rely on memory and may thus be less precise, although in-task assessments could have heightened participants’ awareness of these emotions and biased their decisions. Second, our measures of guilt and shame depend on participants’ conceptual understanding of the two emotions. While this is common practice in studies with adult participants (Michl et al., 2014; Wagner et al., 2011; Zhu et al., 2019), it may be less appropriate for research involving children.”

We apologize for the confusion. The 16 scenarios were presented in a random order. We have clarified this in the revised manuscript (Page 13): “After the interpersonal game, the outcomes of the experimental trials were re-presented in a random order.”

(2) In the neural analysis of emotion sensitivity, the authors identify brain regions correlated with responsibility-driven shame sensitivity and then use those brain regions as masks to test whether they were more involved in the responsibility-driven shame sensitivity than the other types of emotion sensitivity. I wonder if this is biasing the results. Would it be better to use a cross-validation approach? A similar issue might arise in "Activation analysis (neural basis of compensatory sensitivity)."

Thank you for this valuable comment. We replaced the original analyses with a leave-one-subject-out (LOSO) cross-validation approach, which minimizes bias in secondary tests due to non-independence (Esterman et al., 2010). The findings were largely consistent with the original results, except that two previously significant effects became marginally significant (one effect changed from P = 0.012 to P = 0.053; the other from P = 0.044 to P = 0.062). Although we believe the new results do not alter our main conclusions, marginally significant findings should be interpreted with caution. We have noted this point in the Discussion section (Page 48): “… marginally significant results should be viewed cautiously and warrant further examination in future studies with larger sample sizes.”

In the revised manuscript, we have described the cross-validation procedure in detail and reported the corresponding results. Please see the Method section, Page 23: “The results showed that the neural responses in the temporoparietal junction/superior temporal sulcus (TPJ/STS) and precentral cortex/postcentral cortex/supplementary motor area (PRC/POC/SMA) were negatively correlated with the responsibility-driven shame sensitivity. To test whether these regions were more involved in responsibilitydriven shame sensitivity than in other types of emotion sensitivity, we implemented a leave-one-subject-out (LOSO) cross-validation procedure (e.g., Esterman et al., 2010). In each fold, clusters in the TPJ/STS and PRC/POC/SMA showing significant correlations with responsibility-driven shame sensitivity were identified at the group level based on N-1 participants. These clusters, defined as regions of interest (ROI), were then applied to the left-out participant, from whom we extracted the mean parameter estimates (i.e., neural response values). If, in a given fold, no suprathreshold cluster was detected within the TPJ/STS or PRC/POC/SMA after correction, or if the two regions merged into a single cluster that could not be separated, the corresponding value was coded as missing. Repeating this procedure across all folds yielded an independent set of ROI-based estimates for each participant. In the LOSO crossvalidation procedure, the TPJ/STS and PRC/POC/SMA merged into a single inseparable cluster in two folds, and no suprathreshold cluster was detected within the TPJ/STS in one fold. These instances were coded as missing, resulting in valid data from 39 participants for the TPJ/STS and 40 participants for the PRC/POC/SMA. We then correlated these estimates with all four types of emotion sensitivities and compared the correlation with responsibility-driven shame sensitivity against those with the other sensitivities using Z tests (Pearson and Filon's Z).” and Page 24: “To directly test whether these regions were more involved in one of the two types of compensatory sensitivity, we applied the same LOSO cross-validation procedure described above. In this procedure, no suprathreshold cluster was detected within the LPFC in one fold and within the TP in 27 folds. These cases were coded as missing, resulting in valid data from 42 participants for the bilateral IPL, 41 participants for the LPFC, and 15 participants for the TP. The limited sample size for the TP likely reflects that its effect was only marginally above the correction threshold, such that the reduced power in cross-validation often rendered it nonsignificant. Because the sample size for the TP was too small and the results may therefore be unreliable, we did not pursue further analyses for this region. The independent ROI-based estimates were then correlated with both guilt-driven and shame-driven compensatory sensitivities, and the strength of the correlations was compared using Z tests (Pearson and Filon's Z).”

Please see the Results section, Pages 34 and 35: “To assess whether these brain regions were specifically involved in responsibility-driven shame sensitivity, we compared the Pearson correlations between their activity and all types of emotion sensitivities. The results demonstrated the domain specificity of these regions, by revealing that the TPJ/STS cluster had significantly stronger negative responses to responsibility-driven shame sensitivity than to responsibility-driven guilt sensitivity (Z = 2.44, P = 0.015) and harm-driven shame sensitivity (Z = 3.38, P < 0.001), and a marginally stronger negative response to harm-driven guilt sensitivity (Z = 1.87, P = 0.062) (Figure 4C; Supplementary Table 14). In addition, the sensorimotor areas (i.e., precentral cortex (PRC), postcentral cortex (POC), and supplementary motor area (SMA)) exhibited the similar activation pattern as the TPJ/STS (Figure 4B and 4C; Supplementary Tables 13 and 14).” and Page 35: “The results revealed that the left LPFC was more engaged in shame-driven compensatory sensitivity (Z = 1.93, P = 0.053), as its activity showed a marginally stronger positive correlation with shamedriven sensitivity than with guilt-driven sensitivity (Figure 5C). No significant difference was found in the Pearson correlations between the activity of the bilateral IPL and the two types of sensitivities (Supplementary Table 16). For the TP, the effective sample size was too small to yield reliable results (see Methods).”

(1) Regarding the traits of guilt and shame, I appreciate using the scores from the subscales (evaluations and action tendencies) separately for the analyses (instead of a composite score). An issue with using the actions subscales when measuring guilt and shame proneness is that the behavioral tendencies for each emotion get conflated with their definitions, risking circularity. It is reassuring that the behavior evaluation subscale was significantly correlated with compensatory behavior (not only the action tendencies subscale). However, the absence of significant neural correlates for the behavior evaluation subscale raises questions: Do the authors have thoughts on why this might be the case, and any implications?

We are grateful for this important comment. According to the Guilt and Shame Proneness Scale, trait guilt comprises two dimensions: negative behavior evaluations and repair action tendencies (Cohen et al., 2011). Behaviorally, both dimensions were significantly correlated with participants’ compensatory behavior (negative behavior evaluations: R = 0.39, P = 0.010; repair action tendencies: R = 0.33, P = 0.030). Neurally, while repair action tendencies were significantly associated with activity in the aMCC and other brain areas, negative behavior evaluations showed no significant neural correlates. The absence of significant neural correlates for negative behavior evaluations may be due to several factors. In addition to common explanations (e.g., limited sample size reducing the power to detect weak neural correlates or subtle effects obscured by fMRI noise), another possibility is that this dimension influences neural responses indirectly through intermediate processes not captured in our study (e.g., specific motivational states). We have added a discussion of the non-significant result to the revised manuscript (Page 47): “However, the neural correlates of negative behavior evaluations (another dimension of trait guilt) were absent. The reasons underlying the non-significant neural finding may be multifaceted. One possibility is that negative behavior evaluations influence neural responses indirectly through intermediate processes not captured in our study (e.g., specific motivational states).”

In addition, to avoid misunderstanding, the revised manuscript specifies at the appropriate places that the neural findings pertain to repair action tendencies rather than to trait guilt in general. For instance, see Pages 46 and 47: “Furthermore, we found neural responses in the aMCC mediated the relationship between repair action tendencies (one dimension of trait guilt) and compensation… Accordingly, our fMRI findings suggest that individuals with stronger tendency to engage in compensation across various moral violation scenarios (indicated by their repair action tendencies) are more sensitive to the severity of the violation and therefore engage in greater compensatory behavior.”

(2) Regarding the computational model finding that participants seem to disregard selfinterest, do the authors believe it may reflect the relatively small endowment at stake? Do the authors believe this behavior would persist if the stakes were higher?Additionally, might the type of harm inflicted (e.g., electric shock vs. less stigmatized/less ethically charged harm like placing a hand in ice-cold water) influence the weight of self-interest in decision-making?Taken together, the conclusions of the paper are well supported by the data. It would be valuable for future studies to validate these findings using alternative tasks or paradigms to ensure the robustness and generalizability of the observed behavioral and neural mechanisms.

Thank you for these important questions. As you suggested, we believe that the relatively small personal stakes in our task (a maximum loss of 5 Chinese yuan) likely explain why the computational model indicated that participants disregarded selfinterest. We also agree that when the harm to others is less morally charged, people may be more inclined to consider self-interest in compensatory decision-making. Overall, the more stigmatized the harm and the smaller the personal stakes, the more likely individuals are to disregard self-interest and focus solely on making appropriate compensation.

We have added the following passage to the Discussion section (Page 42): “Notably, in many computational models of social decision-making, self-interest plays a crucial role (e.g., Wu et al., 2024). However, our computational findings suggest that participants disregarded self-interest during compensatory decision-making. A possible explanation is that the personal stakes in our task were relatively small (a maximum loss of 5 Chinese yuan), whereas the harm inflicted on the receiver was highly stigmatized (i.e., an electric shock). Under conditions where the harm is highly salient and the cost of compensation is low, participants may be inclined to disregard selfinterest and focus solely on making appropriate compensation.”

**Reviewer #2 (Public review):**
SummaryThe authors combined behavioral experiments, computational modeling, and functional magnetic resonance imaging (fMRI) to investigate the psychological and neural mechanisms underlying guilt, shame, and the altruistic behaviors driven by these emotions. The results revealed that guilt is more strongly associated with harm, whereas shame is more closely linked to responsibility. Compared to shame, guilt elicited a higher level of altruistic behavior. Computational modeling demonstrated how individuals integrate information about harm and responsibility. The fMRI findings identified a set of brain regions involved in representing harm and responsibility, transforming responsibility into feelings of shame, converting guilt and shame into altruistic actions, and mediating the effect of trait guilt on compensatory behavior.StrengthsThis study offers a significant contribution to the literature on social emotions by moving beyond prior research that typically focused on isolated aspects of guilt and shame. The study presents a comprehensive examination of these emotions, encompassing their cognitive antecedents, affective experiences, behavioral consequences, trait-level characteristics, and neural correlates. The authors have introduced a novel experimental task that enables such a systematic investigation and holds strong potential for future research applications. The computational modeling procedures were implemented in accordance with current field standards. The findings are rich and offer meaningful theoretical insights. The manuscript is well written, and the results are clearly and logically presented.

We are thankful for your considerate acknowledgment of our work’s strengths and truly value your positive comments.

We would like to note that, in accordance with the journal’s requirements, we have uploaded both a clean version of the revised manuscript and a version with all modifications highlighted in blue.

WeaknessIn this study, participants' feelings of guilt and shame were assessed retrospectively, after they had completed all altruistic decision-making tasks. This reliance on memorybased self-reports may introduce recall bias, potentially compromising the accuracy of the emotion measurements.

Thank you for this crucial comment. We fully agree that measuring guilt and shame after the task may affect accuracy to some extent. However, because participants reported their emotions immediately after completing the task, we believe their recollections were reasonably accurate. In designing the experiment, we considered intask assessments, but this approach risked heightening participants’ awareness of guilt and shame and thereby interfering with compensatory decisions. After careful consideration, we ultimately chose post-task assessments of these emotions. A similar approach has been adopted in prior research on gratitude, where post-task assessments were also used (Yu et al., 2018).

In the revised manuscript, we have specified the limitations of both post-task and intask assessments of guilt and shame (Page 47): “… post-task assessments of guilt and shame, unlike in-task assessments, rely on memory and may thus be less precise, although in-task assessments could have heightened participants’ awareness of these emotions and biased their decisions.”.

In many behavioral economic models, self-interest plays a central role in shaping individual decision-making, including moral decisions. However, the model comparison results in this study suggest that models without a self-interest component (such as Model 1.3) outperform those that incorporate it (such as Model 1.1 and Model 1.2). The authors have not provided a satisfactory explanation for this counterintuitive finding.

Thank you for this important comment. In the revised manuscript, we have provided a possible explanation (Page 42): “Notably, in many computational models of social decision-making, self-interest plays a crucial role (e.g., Wu et al., 2024). However, our computational findings suggest that participants disregarded self-interest during compensatory decision-making. A possible explanation is that the personal stakes in our task were relatively small (a maximum loss of 5 Chinese yuan), whereas the harm inflicted on the receiver was highly stigmatized (i.e., an electric shock). Under conditions where the harm is highly salient and the cost of compensation is low, participants may be inclined to disregard self-interest and focus solely on making appropriate compensation.”

The phrases "individuals integrate harm and responsibility in the form of a quotient" and "harm and responsibility are integrated in the form of a quotient" appear in the Abstract and Discussion sections. However, based on the results of the computational modeling, it is more accurate to state that "harm and the number of wrongdoers are integrated in the form of a quotient." The current phrasing misleadingly suggests that participants represent information as harm divided by responsibility, which does not align with the modeling results. This potentially confusing expression should be revised for clarity and accuracy.

We sincerely thank you for this helpful suggestion and apologize for the confusion caused. We have removed expressions such as “harm and responsibility are integrated in the form of a quotient” from the manuscript. Instead, we now state more precisely that “harm and the number of wrongdoers are integrated in the form of a quotient.”

However, in certain contexts we continue to discuss harm and responsibility. Introducing “the number of wrongdoers” in these places would appear abrupt, so we have opted for alternative phrasing. For example, on Page 3, we now write:

“Computational modeling results indicated that the integration of harm and responsibility by individuals is consistent with the phenomenon of responsibility diffusion.” Similarly, on Page 49, we state: “Notably, harm and responsibility are integrated in a manner consistent with responsibility diffusion prior to influencing guilt-driven and shame-driven compensation.”

In the Discussion, the authors state: "Since no brain region associated with social cognition showed significant responses to harm or responsibility, it appears that the human brain encodes a unified measure integrating harm and responsibility (i.e., the quotient) rather than processing them as separate entities when both are relevant to subsequent emotional experience and decision-making." However, this interpretation overstates the implications of the null fMRI findings. The absence of significant activation in response to harm or responsibility does not necessarily imply that the brain does not represent these dimensions separately. Null results can arise from various factors, including limitations in the sensitivity of fMRI. It is possible that more finegrained techniques, such as intracranial electrophysiological recordings, could reveal distinct neural representations of harm and responsibility. The interpretation of these null findings should be made with greater caution.

Thank you for this reminder. In the revised manuscript, we have provided a more cautious interpretation of the results (Page 43): “Although the fMRI findings revealed that no brain region associated with social cognition showed significant responses to harm or responsibility, this does not suggest that the human brain encodes only a unified measure integrating harm and responsibility and does not process them as separate entities. Using more fine-grained techniques, such as intracranial electrophysiological recordings, it may still be possible to observe independent neural representations of harm and responsibility.”

**Reviewer #3 (Public review):**
SummaryZhu et al. set out to elucidate how the moral emotions of guilt and shame emerge from specific cognitive antecedents - harm and responsibility - and how these emotions subsequently drive compensatory behavior. Consistent with their prediction derived from functionalist theories of emotion, their behavioral findings indicate that guilt is more influenced by harm, whereas shame is more influenced by responsibility. In line with previous research, their results also demonstrate that guilt has a stronger facilitating effect on compensatory behavior than shame. Furthermore, computational modeling and neuroimaging results suggest that individuals integrate harm and responsibility information into a composite representation of the individual's share of the harm caused. Brain areas such as the striatum, insula, temporoparietal junction, lateral prefrontal cortex, and cingulate cortex were implicated in distinct stages of the processing of guilt and/or shame. In general, this work makes an important contribution to the field of moral emotions. Its impact could be further enhanced by clarifying methodological details, offering a more nuanced interpretation of the findings, and discussing their potential practical implications in greater depth.StrengthsFirst, this work conceptualizes guilt and shame as processes unfolding across distinct stages (cognitive appraisal, emotional experience, and behavioral response) and investigates the psychological and neural characteristics associated with their transitions from one stage to the next.Second, the well-designed experiment effectively manipulates harm and responsibility - two critical antecedents of guilt and shame.Third, the findings deepen our understanding of the mechanisms underlying guilt and shame beyond what has been established in previous research.

We truly appreciate your acknowledgment of our work’s strengths and your encouraging feedback.

We would like to note that, in accordance with the journal’s requirements, we have uploaded both a clean version of the revised manuscript and a version with all modifications highlighted in blue.

WeaknessOver the course of the task, participants may gradually become aware of their high error rate in the dot estimation task. This could lead them to discount their own judgments and become inclined to rely on the choices of other deciders. It is unclear whether participants in the experiment had the opportunity to observe or inquire about others' choices. This point is important, as the compensatory decision-making process may differ depending on whether choices are made independently or influenced by external input.

Thank you for pointing this out. We apologize for not making the experimental procedure sufficiently clear. Participants (as deciders) were informed that each decider performed the dot estimation independently and was unaware of the estimations made by the other deciders. We now have clarified this point in the revised manuscript (Pages 10 and 11): “Each decider indicated whether the number of dots was more than or less than 20 based on their own estimation by pressing a corresponding button (dots estimation period, < 2.5 s) and was unaware of the estimations made by other deciders”.

Given the inherent complexity of human decision-making, it is crucial to acknowledge that, although the authors compared eight candidate models, other plausible alternatives may exist. As such, caution is warranted when interpreting the computational modeling results.

Thank you for this comment. We fully agree with your opinion. Although we tried to build a conceptually comprehensive model space based on prior research and our own understanding, we did not include all plausible models, nor would it be feasible to do so. We acknowledge it as a limitation in the revised manuscript (Page 47): “... although we aimed to construct a conceptually comprehensive computational model space informed by prior research and our own understanding, it does not encompass all plausible models. Future research is encouraged to explore additional possibilities.”

I do not agree with the authors' claim that "computational modeling results indicated that individuals integrate harm and responsibility in the form of a quotient" (i.e., harm/responsibility). Rather, the findings appear to suggest that individuals may form a composite representation of the harm attributable to each individual (i.e., harm/the number of people involved). The explanation of the modeling results ought to be precise.

We appreciate your comment and apologize for the imprecise description. In the revised manuscript, we now use the expressions “… integrate harm and the number of wrongdoers in the form of a quotient.” and “… the integration of harm and responsibility by individuals is consistent with the phenomenon of responsibility diffusion.” For example, on Page 19, we state: “It assumes that individuals neglect their self-interest, have a compensatory baseline, and integrate harm and the number of wrongdoers in the form of a quotient.” On Page 3, we state: “Computational modeling results indicated that the integration of harm and responsibility by individuals is consistent with the phenomenon of responsibility diffusion.”

Many studies have reported positive associations between trait gratitude, social value orientation, and altruistic behavior. It would be helpful if the authors could provide an explanation about why this study failed to replicate these associations.

Thanks a lot for this important comment. We have now added an explanation into the revised manuscript (Page 47): “Although previous research has found that trait gratitude and SVO are significantly associated with altruistic behavior in contexts such as donation (Van Lange et al., 2007; Yost-Dubrow & Dunham, 2018) and reciprocity (Ma et al., 2017; Yost-Dubrow & Dunham, 2018), their associations with compensatory decisions in the present study were not significant. This suggests that the effects of trait gratitude and SVO on altruistic behavior are context-dependent and may not predict all forms of altruistic behavior.”

As the authors noted, guilt and shame are closely linked to various psychiatric disorders. It would be valuable to discuss whether this study has any implications for understanding or even informing the treatment of these disorders.

We are grateful for this advice. Although our study did not directly examine patients with psychological disorders, the findings offer insights into the regulation of guilt and shame. As these emotions are closely linked to various disorders, improving their regulation may help alleviate related symptoms. Accordingly, we have added a paragraph highlighting the potential clinical relevance (Pages 48 and 49): “Our study has potential practical implications. The behavioral findings may help counselors understand how cognitive interventions targeting perceptions of harm and responsibility could influence experiences of guilt and shame. The neural findings highlight specific brain regions (e.g., TPJ) as potential intervention targets for regulating these emotions. Given the close links between guilt, shame, and various psychological disorders (e.g., Kim et al., 2011; Lee et al., 2001; Schuster et al., 2021), strategies to regulate these emotions may contribute to symptom alleviation. Nevertheless, because this study was conducted with healthy adults, caution is warranted when considering applications to other populations.”

**Reviewer #1 (Recommendations for the authors):**
(1) Would it be interesting to explore other categories of behavior apart from compensatory behavior?

Thanks a lot for this insightful question. We focused on a classic form of altruistic behavior, compensation. Future studies are encouraged to adapt our paradigm to examine other behaviors associated with guilt and/or shame, such as donation (Xu, 2022), avoidance (Shen et al., 2023), or aggression (Velotti et al., 2014). Please see Page 48: “Future research could combine this paradigm with other cognitive neuroscience methods, such as electroencephalography (EEG) or magnetoencephalography (MEG), and adapt it to investigate additional behaviors linked to guilt and shame, including donation (Xu, 2022), avoidance (Shen et al., 2023), and aggression (Velotti et al., 2014).”

(2) Did the computational model account for the position of the block (slider) at the start of each decision-making response (when participants had to decide how to divide the endowment)? Or are anchoring effects not relevant/ not a concern?

Thank you for this interesting question. In our task, the initial position of the slider was randomized across trials, and participants were explicitly informed of this in the instructions. This design minimized stable anchoring effects across trials, as participants could not rely on a consistent starting point. Although anchoring might still have influenced individual trial responses, we believe it is unlikely that such effects systematically biased our results, since randomization would tend to cancel them out across trials. Additionally, prior research has shown that when multiple anchors are presented, anchoring effects are reduced if the anchors contradict each other (Switzer III & Sniezek, 1991). Therefore, we did not attempt to model potential anchoring effects. Nevertheless, future research could systematically manipulate slider starting positions to directly examine possible anchoring influences. In the revised manuscript, we have added a brief clarification (Page 11): “The initial position of the block was randomized across trials, which helped minimize stable anchoring effects across trials.”

(3) Was there a real receiver who experienced the shocks and received compensation? I think it is not completely clear in the paper.

We are sorry for not making this clear enough. The receiver was fictitious and did not actually exist. We have supplemented the Methods section with the following description (Page 12): “We told the participant a cover story that the receiver was played by another college student who was not present in the laboratory at the time. … In fact, the receiver did not actually exist.”.

(4) What was the rationale behind not having participants meet the receiver?

Thank you for this question. Having participants meet the receiver (i.e., the victim), played by a confederate, might have intensified their guilt and shame and produced a ceiling effect. In addition, the current approach simplified the experimental procedure and removed the need to recruit an additional confederate. These reasons have been added to the Methods section (Page 12): “Not having participants meet the receiver helped prevent excessive guilt and shame that might produce a ceiling effect, while also eliminating the need to recruit an additional confederate.”

Minor edits:(1) Line 49: "the cognitive assessment triggers them", I think a word is missing.(2) Line 227: says 'Slide' instead of 'Slider'.(3) Lines 867/868: "No brain response showed significant correlation with responsibility-driven guilt sensitivity, harm-driven shame sensitivity, or responsibilitydriven shame sensitivity." I think it should be harm-driven guilt sensitivity, responsibility-driven guilt sensitivity, and harm-driven shame sensitivity.(4) Supplementary Information Line 12: I think there is a typo ('severs' instead of 'serves')

We sincerely thank you for patiently pointing out these typos. We have corrected them accordingly.

(1) “the cognitive assessment triggers them” has been revised to “the cognitive antecedents that trigger them” (Page 2).

(2) “SVO Slide Measure” has been revised to “SVO Slider Measure” (Page 8).

(3) “No brain response showed significant correlation with responsibility-driven guilt sensitivity, harm-driven shame sensitivity, or responsibility-driven shame sensitivity." has been revised to “No brain response showed significant correlation with harm-driven guilt sensitivity, responsibility-driven guilt sensitivity, and harm-driven shame sensitivity.” (Page 35).

(4) “severs” has been revised to “serves” (see Supplementary Information). In addition, we have carefully checked the entire manuscript to correct any remaining typographical errors.

**Reviewer #2 (Recommendations for the authors):**
The statement that trait gratitude and SVO were measured "for exploratory purposes" would benefit from further clarification regarding the specific questions being explored.

Thank you for this valuable suggestion. In the revised manuscript, we have illustrated the exploratory purposes (Page 9): “We measured trait gratitude and SVO for exploratory purposes. Previous research has shown that both are linked to altruistic behavior, particularly in donation contexts (Van Lange et al., 2007; Yost-Dubrow & Dunham, 2018) and reciprocity contexts (Ma et al., 2017; Yost-Dubrow & Dunham, 2018). Here, we explored whether they also exert significant effects in a compensatory context.”

In the Methods section, the authors state: "To confirm the relationships between κ and guilt-driven and shame-driven compensatory sensitivities, we calculated the Pearson correlations between them." However, the Results section reports linear regression results rather than Pearson correlation coefficients, suggesting a possible inconsistency. The authors are advised to carefully check and clarify the analysis approach used.

We thank you for the careful reviewing and apologize for this mistake. We used a linear mixed-effects regression instead of Pearson correlations for the analysis. The mistake has been revised (Page 25): “To confirm the relationships between κ and guiltdriven and shame-driven compensatory sensitivities, we conducted a linear mixedeffects regression. κ was regressed onto guilt-driven and shame-driven compensatory sensitivities, with participant-specific random intercepts and random slopes for each fixed effect included as random effects.”

A more detailed discussion of how the current findings inform the regulation of guilt and shame would further strengthen the contribution of this study.

Thank you for this suggestion. We have added a paragraph discussing the implications for the regulation of guilt and shame (Pages 48 and 49): “Our study has potential practical implications. The behavioral findings may help counselors understand how cognitive interventions targeting perceptions of harm and responsibility could influence experiences of guilt and shame. The neural findings highlight specific brain regions (e.g., TPJ) as potential intervention targets for regulating these emotions. Given the close links between guilt, shame, and various psychological disorders (e.g., Kim et al., 2011; Lee et al., 2001; Schuster et al., 2021), strategies to regulate these emotions may contribute to symptom alleviation. Nevertheless, because this study was conducted with healthy adults, caution is warranted when considering applications to other populations.”

As fMRI provides only correlational evidence, establishing a causal link between neural activity and guilt- or shame-related cognition and behavior would require brain stimulation or other intervention-based methods. This may represent a promising direction for future research.

Thank you for this advice. We also agree that it is important for future research to establish the causal relationships between the observed brain activity, psychological processes, and behavior. We have added a corresponding discussion in the revised manuscript (Pages 47 and 48): “… fMRI cannot establish causality. Future studies using brain stimulation techniques (e.g., transcranial magnetic stimulation) are needed to clarify the causal role of brain regions in guilt-driven and shame-driven altruistic behavior.”

**Reviewer #3 (Recommendations for the authors):**
It was mentioned that emotions beyond guilt and shame, such as indebtedness, may also drive compensation. Were any additional types of emotion measured in the study?

Thank you for this question. We did not explicitly measure emotions other than guilt and shame. However, the parameter κ from our winning computational model captures the combined influence of various psychological processes on compensation, which may reflect the impact of emotions beyond guilt and shame (e.g., indebtedness). We acknowledge that measuring other emotions similar to guilt and shame may help to better understand their distinct contributions. This point has been added into the revised manuscript (Page 48): “… we did not explicitly measure emotions similar to guilt and shame (e.g., indebtedness), which would have been helpful for understanding their distinct contributions.”

The experimental task is complicated, raising the question of whether participants fully understood the instructions. For instance, one participant's compensation amount was zero. Could this reflect a misunderstanding of the task instructions?

Thanks a lot for this question. In our study, after reading the instructions, participants were required to complete a comprehension test on the experimental rules. If they made any mistakes, the experimenter provided additional explanations. Only after participants fully understood the rules and correctly answered all comprehension questions did they proceed to the main experimental task. We have clarified this procedure in the revised manuscript (Page 13): “Participants did not proceed to the interpersonal game until they had fully understood the experimental rules and passed a comprehension test.”

Making identical choices across different trials does not necessarily indicate that participants misunderstood the rules. Similar patterns, where participants made the same choices across trials, have also been observed in previous studies (Zhong et al., 2016; Zhu et al., 2021).

Reference

Cohen, T. R., Wolf, S. T., Panter, A. T., & Insko, C. A. (2011). Introducing the GASP scale: a new measure of guilt and shame proneness. Journal of Personality and Social Psychology, 100(5), 947–966. https://doi.org/10.1037/a0022641

Esterman, M., Tamber-Rosenau, B. J., Chiu, Y. C., & Yantis, S. (2010). Avoiding nonindependence in fMRI data analysis: Leave one subject out. NeuroImage, 50(2), 572–576. https://doi.org/10.1016/j.neuroimage.2009.10.092

Kim, S., Thibodeau, R., & Jorgensen, R. S. (2011). Shame, guilt, and depressive symptoms: A meta-analytic review. Psychological Bulletin, 137(1), 68. https://doi.org/10.1037/a0021466

Lee, D. A., Scragg, P., & Turner, S. (2001). The role of shame and guilt in traumatic events: A clinical model of shame-based and guilt-based PTSD. British Journal of Medical Psychology, 74(4), 451–466. https://doi.org/10.1348/000711201161109

Ma, L. K., Tunney, R. J., & Ferguson, E. (2017). Does gratitude enhance prosociality?: A meta-analytic review. Psychological Bulletin, 143(6), 601–635. https://doi.org/10.1037/bul0000103

Michl, P., Meindl, T., Meister, F., Born, C., Engel, R. R., Reiser, M., & Hennig-Fast, K. (2014). Neurobiological underpinnings of shame and guilt: A pilot fMRI study. Social Cognitive and Affective Neuroscience, 9(2), 150–157.

Schuster, P., Beutel, M. E., Hoyer, J., Leibing, E., Nolting, B., Salzer, S., Strauss, B., Wiltink, J., Steinert, C., & Leichsenring, F. (2021). The role of shame and guilt in social anxiety disorder. Journal of Affective Disorders Reports, 6, 100208. https://doi.org/10.1016/j.jadr.2021.100208

Shen, B., Chen, Y., He, Z., Li, W., Yu, H., & Zhou, X. (2023). The competition dynamics of approach and avoidance motivations following interpersonal transgression. Proceedings of the National Academy of Sciences, 120(40), e2302484120. https://doi.org/10.1073/pnas.230248412

Switzer III, F. S., & Sniezek, J. A. (1991). Judgment processes in motivation: Anchoring and adjustment effects on judgment and behavior. Organizational Behavior and Human Decision Processes, 49(2), 208–229. https://doi.org/10.1016/0749-5978(91)90049-Y

Van Lange, P. A. M., Bekkers, R., Schuyt, T. N. M., & Van Vugt, M. (2007). From games to giving: Social value orientation predicts donations to noble causes. Basic and Applied Social Psychology, 29(4), 375–384. https://doi.org/10.1080/01973530701665223

Velotti, P., Elison, J., & Garofalo, C. (2014). Shame and aggression: Different trajectories and implications. Aggression and Violent Behavior, 19(4), 454–461. https://doi.org/10.1016/j.avb.2014.04.011

Wagner, U., N’Diaye, K., Ethofer, T., & Vuilleumier, P. (2011). Guilt-specific processing in the prefrontal cortex. Cerebral Cortex, 21(11), 2461–2470. https://doi.org/10.1093/cercor/bhr016

Wu, X., Ren, X., Liu, C., & Zhang, H. (2024). The motive cocktail in altruistic behaviors. Nature Computational Science, 4, 659–676. https://doi.org/10.1038/s43588-024-00685-6

Xu, J. (2022). The impact of guilt and shame in charity advertising: The role of self- construal. Journal of Philanthropy and Marketing, 27(1). https://doi.org/10.1002/nvsm.1709

Yost-Dubrow, R., & Dunham, Y. (2018). Evidence for a relationship between trait gratitude and prosocial behaviour. Cognition and Emotion, 32(2), 397–403. https://doi.org/10.1080/02699931.2017.1289153

Yu, H., Gao, X., Zhou, Y., & Zhou, X. (2018). Decomposing gratitude: Representation and integration of cognitive antecedents of gratitude in the brain. Journal of Neuroscience, 38(21), 4886–4898. https://doi.org/10.1523/JNEUROSCI.2944-17.2018

Zhong, S., Chark, R., Hsu, M., & Chew, S. H. (2016). Computational substrates of social norm enforcement by unaffected third parties. NeuroImage, 129, 95–104. https://doi.org/10.1016/j.neuroimage.2016.01.040

Zhu, R., Feng, C., Zhang, S., Mai, X., & Liu, C. (2019). Differentiating guilt and shame in an interpersonal context with univariate activation and multivariate pattern analyses. NeuroImage, 186, 476486. https://doi.org/10.1016/j.neuroimage.2018.11.012

Zhu, R., Xu, Z., Su, S., Feng, C., Luo, Y., Tang, H., Zhang, S., Wu, X., Mai, X., & Liu, C. (2021). From gratitude to injustice: Neurocomputational mechanisms of gratitude-induced injustice. NeuroImage, 245, 118730. https://doi.org/10.1016/j.neuroimage.2021.118730